# Deciphering the Genomic Landscape and Virulence Mechanisms of the Wheat Powdery Mildew Pathogen *Blumeria graminis* f. sp. *tritici* Wtn1: Insights from Integrated Genome Assembly and Conidial Transcriptomics

**DOI:** 10.3390/jof10040267

**Published:** 2024-04-03

**Authors:** Perumal Nallathambi, Chandrasekaran Umamaheswari, Bhaskar Reddy, Balakrishnan Aarthy, Mohammed Javed, Priya Ravikumar, Santosh Watpade, Prem Lal Kashyap, Govindaraju Boopalakrishnan, Sudheer Kumar, Anju Sharma, Aundy Kumar

**Affiliations:** 1ICAR-Indian Agricultural Research Institute, Regional Station, Wellington 643231, Tamil Nadu, India; nallathambiiari@gmail.com (P.N.); umaiari.24@gmail.com (C.U.); aarthybalakrishnan94@gmail.com (B.A.); priyaravikumar25@gmail.com (P.R.); 2ICAR-Indian Agricultural Research Institute, Pusa Campus, New Delhi 110012, Delhi, India; jvd.choudhary@gmail.com (M.J.); boobalg1985@gmail.com (G.B.); 3ICAR-Indian Agricultural Research Institute, Regional Station, Shimla 171004, Himachal Pradesh, India; santoshpathology@gmail.com; 4ICAR-Indian Institute of Wheat and Barley Research, Karnal 132001, Haryana, India; plkashyap@gmail.com (P.L.K.); sudheer.kumar@icar.gov.in (S.K.); anjusharma20april@gmail.com (A.S.)

**Keywords:** *Blumeria*, fungi, genome, powdery mildew, transcriptome, wheat

## Abstract

A high-quality genome sequence from an Indian isolate of *Blumeria graminis* f. sp. *tritici* Wtn1, a persistent threat in wheat farming, was obtained using a hybrid method. The assembly of over 9.24 million DNA-sequence reads resulted in 93 contigs, totaling a 140.61 Mb genome size, potentially encoding 8480 genes. Notably, more than 73.80% of the genome, spanning approximately 102.14 Mb, comprises retro-elements, LTR elements, and P elements, influencing evolution and adaptation significantly. The phylogenomic analysis placed *B. graminis* f. sp. *tritici* Wtn1 in a distinct monocot-infecting clade. A total of 583 tRNA anticodon sequences were identified from the whole genome of the native virulent strain *B. graminis* f. sp. *tritici*, which comprises distinct genome features with high counts of tRNA anticodons for leucine (70), cysteine (61), alanine (58), and arginine (45), with only two stop codons (Opal and Ochre) present and the absence of the Amber stop codon. Comparative InterProScan analysis unveiled “shared and unique” proteins in *B. graminis* f. sp. *tritici* Wtn1. Identified were 7707 protein-encoding genes, annotated to different categories such as 805 effectors, 156 CAZymes, 6102 orthologous proteins, and 3180 distinct protein families (PFAMs). Among the effectors, genes like *Avra10*, *Avrk1*, *Bcg-7*, *BEC1005*, *CSEP0105*, *CSEP0162*, *BEC1016*, *BEC1040*, and *HopI1* closely linked to pathogenesis and virulence were recognized. Transcriptome analysis highlighted abundant proteins associated with RNA processing and modification, post-translational modification, protein turnover, chaperones, and signal transduction. Examining the Environmental Information Processing Pathways in *B. graminis* f. sp. *tritici* Wtn1 revealed 393 genes across 33 signal transduction pathways. The key pathways included yeast MAPK signaling (53 genes), mTOR signaling (38 genes), PI3K-Akt signaling (23 genes), and AMPK signaling (21 genes). Additionally, pathways like FoxO, Phosphatidylinositol, the two-component system, and Ras signaling showed significant gene representation, each with 15–16 genes, key SNPs, and Indels in specific chromosomes highlighting their relevance to environmental responses and pathotype evolution. The SNP and InDel analysis resulted in about 3.56 million variants, including 3.45 million SNPs, 5050 insertions, and 5651 deletions within the whole genome of *B. graminis* f. sp. *tritici* Wtn1. These comprehensive genome and transcriptome datasets serve as crucial resources for understanding the pathogenicity, virulence effectors, retro-elements, and evolutionary origins of *B. graminis* f. sp. *tritici* Wtn1, aiding in developing robust strategies for the effective management of wheat powdery mildew.

## 1. Introduction

Wheat is one of the most extensively farmed and consumed cereal grains globally, serving as a crucial dietary cornerstone for millions. Its adaptability to diverse climates renders it a dependable food staple across numerous nations, contributing significantly to global food security [1]. Yet, innumerable diseases, including rusts, blights, and smuts, threaten wheat crops capable of diminishing yield and quality, thereby causing substantial economic setbacks for farmers. Among these, rust and smuts have been familiar to humanity since ancient times. However, another disease, powdery mildew, ranks among the top ten diseases that severely curtail wheat productivity [2]. The resurgence of this powdery mildew caused by *B. graminis* f. sp. *tritici* is becoming evident across numerous wheat-growing nations, resulting in a yield loss ranging from 10 to 62% in countries such as Russia, South America, the United Kingdom, Europe, Brazil, Canada, and China [3]. Within India, the projected yield reduction ranges from 13–34% for moderate infections to a staggering 50–100% during severe outbreaks. A sole infection by a solitary conidium, leading to the formation of a mildew colony, releases around two million conidia capable of traveling up to 650 km that exhibit resistance not only to cold conditions but also drought environments [4,5].

Repetitive infections inevitably result in new pathotypes capable of breaking the host resistance [6,7]. *B. graminis* f. sp. *tritici* also forms sexual ascospores within the chasmothecia on mature wheat leaves, aiding the fungus in surviving summer temperatures during the offseason [8]. Upon invading the wheat host, the fungus develops lobed haustoria, either to absorb nutrients or trigger effector-mediated immune signaling if the functional immune receptor is present [9,10,11,12]. The haustorium additionally provides an expansive surface for exchanging proteins like Candidate Secreted Effector Protein (CSEP) and metabolites between the pathogen and the host [12,13]. These proteins are capable of manipulating the host cell machinery to trigger susceptibility by suppressing basal immune responses, aiding fungal growth and reproduction [14,15].

Utilizing molecular methods to analyze genomic data can aid in identifying various forma specialis, molecular phylogenetics, population biology, genetics, and evolutionary origins [16,17]. Previously, *Blumeria* infecting wheat (*B. graminis* f. sp. *tritici*) and barley (*B. graminis* f. sp. *hordei*) was sequenced using the Sanger method [18]. However, the assemblies were fragmented and of moderate quality [18,19]. In contrast to Sanger’s method, NGS technologies offer robust genome coverage and depth, enabling the exploration of genomic capabilities. For example, recent genome sequencing has revealed a wide array of effectors within powdery mildew species infecting dicots and monocots [18,19,20,21,22,23].

Additionally, genomic data can assist in the discovery of novel powdery mildew resistance (*Pm*) genes that could be incorporated into hexaploid wheat via breeding initiatives. However, sequencing the *B. graminis* f. sp. *tritici* genome has consistently posed a formidable challenge [24]. Its rapid mutation rates, intricate population structure, and varied physiological specialization contribute to the complexity of genome sequencing endeavors [25]. Prior to our study, no documented genome sequences of wheat powdery mildew were published from India. Hence, we aimed to establish genomic resources by sequencing its genome. Employing a hybrid approach, we assembled short reads from Illumina and long genomic reads from Nanopore Technology (ONT), Oxford, UK. Our study had the following primary objectives: (i) isolation of high-quality, high-molecular-weight genomic DNA for library preparation, (ii) whole-genome sequencing using a hybrid assembly to predict genes and gene families, (iii) Gene Ontology analysis and categorization of identified genes, and (iv) unveiling the transcriptome architecture of a virulent type of *B. graminis* f. sp. *tritici* found in India. The genome and transcriptome data generated in our work will support ongoing research endeavors aimed at detecting races/pathotypes across diverse agroclimatic zones and facilitate the implementation of resistance genes tailored to specific zones, enabling effective and sustainable management of wheat powdery mildew.

## 2. Materials and Methods

### 2.1. Virulence Profiling and Selection of the Isolate

Fifteen *B. graminis* f. sp. *tritici* isolates were examined for their virulence profiles. The isolates *B. graminis* f. sp. *tritici* U13, *B. graminis* f. sp. *tritici* Wpm4, *B. graminis* f. sp. *tritici* 17, *B. graminis* f. sp. *tritici* 25, and *B. graminis* f. sp. *tritici* 29 (referred to as *B. graminis* f. sp. *tritici* Wtn1) were collected from the Nilgiris hills in SHZ (Southern Hill Zone). The *B. graminis* f. sp. *tritici* samples SH1, SH4, SH9, SH11, and SH16 were obtained from Shimla in the NHZ (Northern Hill Zone), Himachal Pradesh. Furthermore, the *B. graminis* f. sp. *tritici* samples KPm6, KPm10, KPm21, KPm38, and KPm50 were isolated from Karnal in the NWPZ (North Western Plain Zone of India). These regions represent three distinct agroclimatic zones of India where wheat powdery mildew is highly prevalent (Appendix A).

Wheat seedlings (cv. WL711) were initially used to isolate and purify *B. graminis* f. sp. *tritici* Wtn1. Susceptible seedlings of *Triticum aestivum* (cv. WL711) were employed for this purpose. Additionally, fifty wheat lines, including near-isogenic lines (NILs) with distinct *Pm* (~Powdery Mildew) genes, were inoculated with seven-day-old conidia from fresh colonies. The virulence of the *B. graminis* f. sp. *tritici* Wtn1 isolates was evaluated by observing susceptible reactions on a scale of 0–9 after 7–10 days of powdery mildew inoculations under controlled conditions. A *B. graminis* f. sp. *tritici* isolate was classified as highly virulent if it exhibited a susceptible reaction rating between 7 and 9 on the 0–9 scale. The differential reactions of the *B. graminis* f. sp. *tritici* Wtn1 isolates were identified based on their average scoring values following the approach suggested by Namuco et al. (1987). The total number of virulent isolates and their virulence percentages, varietal efficacies, and *Pm* gene efficacies were computed using resistant and susceptible reactions of Indian wheat varieties and NILs. The calculation of these parameters was conducted according to the formula given by Green (1966).

Additionally, seven-day-old conidia from fresh colonies were introduced into five near-isogenic lines of *Triticum aestivum*, each carrying distinct *Pm* genes like Amigo (*Pm*17), Chul Bidai (*Pm*3b), and Timgalen (*Pm*6). Symptoms and their severity in the seedlings were assessed to determine the virulence (severity and frequencies), varietal resistance performance, and *Pm* gene effectiveness. Virulent reactions were quantified after 7–10 days of conidial inoculation using the 0–9 scale proposed by Namuco et al. (1987). Selection of the virulent strain, *B. graminis* f. sp. *tritici* Wtn1, was based on the frequency of virulence (Appendix A).
$$\mathrm{Virulence}\;\left(\%\right)=\frac{\mathrm{No}.\;\mathrm{of}\;\mathrm{virulent}\;\mathrm{isolates}}{\begin{array}{c} \mathrm{Total}\;\mathrm{number}\;\mathrm{of}\;\mathrm{isolates}\; \end{array}}\times 100$$
$$\mathrm{Varietal}\;\mathrm{resistance}\;\mathrm{efficacy}\;\left(\%\right)=\frac{\mathrm{Number}\;\mathrm{of}\;\mathrm{times}\;\mathrm{the}\;\mathrm{variety}\;\mathrm{is}\;\mathrm{resistant}}{\mathrm{Total}\;\mathrm{number}\;\mathrm{varieties}\;\mathrm{tested}}\times 100$$

Principal Coordinate Analysis (PCA) and neighbor joining clustering, employing correlation as the similarity index were performed to identify the most aggressive and distinctive *B. graminis* f. sp. *tritici* isolates across three representative agro-climatic zones of Indian wheat cultivation.

### 2.2. Selection of the Powdery Mildew Pathogen for WGS

The highly virulent fungal strain, *B. graminis* f. sp. *tritici* Wtn1, used in this study was obtained from a susceptible bread wheat variety (cv. WL 711) collected from the experimental field at ICAR—Indian Agricultural Research Institute, Regional Station, Wellington, situated in the Western Ghats of India (11°21′50.00″ N 76°47′6.90″ E; 1815 Mean Sea Level). Following the validation of its identity and mycological characteristics, pure cultures were maintained on the seedlings using an efficient mass culturing technique. The isolation involved placing the conidia on surface-sterilized leaf segments on a water agar medium supplemented with benzimidazole (60 μg/mL) for genome sequencing (Figure 1).

### 2.3. The Wheat Cultivar and Axenic Culturing of the Powdery Mildew Pathogen

Healthy seedlings emerged from surface-disinfected seeds of the susceptible cv. WL711 were grown for 7–10 days in garden soil (three parts) mixed with decomposed farmyard manure (one part). The autoclaved soil was then placed into sterilized paper cups. Surface-sterilized WL711 seeds were planted in these cups, watered every 48 h with sterile water, and maintained in controlled conditions within a polyhouse (25 ± 1 °C, 85–90% humidity, 16 h light/8 h dark) (Figure 2). For axenic culturing of the *B. graminis* f. sp. *tritici* Wtn1 isolate, primary leaves from the same wheat cultivar were sterilized with ethyl alcohol (70%), cut into 25 mm lengths, and placed on a water agar medium. Fresh conidia of the pathogen were spread-inoculated into these leaves and incubated at 25 ± 2 °C. After seven days, mildew colonies comprising epiphytic conidia were collected under aseptic conditions using a sterilized lancet needle within a laminar flow chamber. Additional leaves were concurrently inoculated to enhance the fungal conidial biomass production (Figure 2).

### 2.4. Mass Production of Conidia and Microscopy

Leaf segments with abundant conidiogenesis of *B. graminis* f. sp *tritici* Wtn1 were picked from a water agar medium, and the conidia on a mildew colony (3 mm^2^) were counted using a hemocytometer. Approximately 25 × 10^3^ conidia per ml were recorded in a suspension prepared from the mildew-infected leaf area (~3.0 mm^2^). For inoculation, 95–100 conidia were placed onto a primary leaf (Length 70–75 mm × Width 4–5 mm) of the susceptible wheat cultivar WL711 to ensure uniform infection. Following inoculation, the seedlings were maintained under aseptic conditions inside incubation chambers made of 2 mm thick translucent polycarbonate sheets. In a controlled playhouse environment, these pots were placed in inoculation cabins (Length 6′× Width 6′× Height 8′). Irrigation (50 mL per pot) was carried out every alternate day via sprinkling. The cabins were illuminated at 100 ± 5 µmol m^−2^ s^−1^ for 16 h of light and 8 h of darkness. After seven days, the *B. graminis* f. sp. *tritici* Wtn1 conidial masses were collected from well-established individual colonies, pooled in sterilized microcentrifuge tubes (2.0 mL each), and preserved at −20 °C for genomic DNA isolation. For mycological observations, fresh conidia were retrieved using a stainless steel nichrome loop (Metaloop^TM^ with a 2 mm diameter), spread on microscopic slides, and examined at different magnifications under a light microscope (Nikon, E600 Eclipse model) equipped with a photomicrography system. The images were visualized at 100× magnification using specific eyepieces (10×/22 Nikon) and objective lenses (10×/0.25 WD 6.1).

### 2.5. Genomic DNA Isolation and Strain Identification

Taxonomic classification at the intra-forma specialis level was performed before the genome sequencing. Initially, efforts were made to purify genetically pure mildew colonies from the infected leaves for genomic DNA extraction (Figure 1), and seven methods using CTAB, SDS, or acombination of both were attempted (Appendix A). Ultimately, a method involving conidia lysis in liquid nitrogen followed by maceration using glass beads in 300 µL of preheated 5% Sarcosyl in CTAB Buffer at 65 °C for 30 min yielded high-molecular-weight genomic DNA. This purified genomic DNA was used as a template for amplifying the fungal species barcodes via a PCR reaction with the ITS1 and ITS4 primers. The resulting PCR amplicons covering the entire barcode length were sequenced bidirectionally using the dideoxy chain termination method. The obtained sequences were end-trimmed, curated, and analyzed using the Basic Local Alignment Search Tool-Nucleotide (BLASTN) to confirm the species identity [26]. After verifying the sequence, it was submitted to the NCBI database and the accession number (MN872365.1) was received.

### 2.6. Amplification of the Pathogenesis-Related Genes

Initially, the PCR primers recommended [27] for Protein Kinase C genes (PKC1 and PKC-like; 714 bp), the Catalase (Cat1; 637 bp) gene, the Alternative Oxidase Gene (159 bp), and the Integral Membrane Protein gene(pth; 735 bp) were used. The PCR mixture at a final volume of 50 µL consisted of 25 µL of 2X DreamTaq PCR Master Mix (Thermo Scientific, Waltham, MA, USA), 1 µL each of the forward and reverse primers (10 pmol/reaction), 4 µL of 25 ng of DNA, and 19 µL of nuclease-free water was subjected to initial denaturation at 95 °C for 3 min; 35 cycles of 30 s at 95 °C, 30 s at 55 °C, and 1 min at 72 °C; followed by a final extension of 12 min at 72 °C. The PCR-amplified products were visualized in 1.0% (*w*/*v*) agarose gel using a 100 bp DNA ladder. The amplified products were sequenced. The sequences were analyzed and accessioned using the NCBI database.

### 2.7. Genomic Libraries and Whole-Genome Sequencing

Genomic libraries for the short and long nucleotide reads were prepared following the manufacturer protocols for genome sequencing (Illumina HiSeq 2500 and PromethION Flow Cell, R9.4.1, Nanopore). The Illumina HiSeq 2500 method employed the NEBNext Ultra DNA Library Preparation Kit, with the subsequent quality assessment conducted using the Agilent TapeStation. In the Nanopore approach, native barcoding of the genomic DNA (EXP-NBD114 and SQK-LSk109) was explicitly executed for the PromethION system. Post-barcoding, stringent quality checks were performed using Qubit and the Agilent TapeStation to ensure the reliability and quality of the prepared libraries.

### 2.8. Sequence Curation and *De Novo* Genome Assembly

The raw fastQ files obtained using Illumina and ONT were subject to comprehensive quality checks, evaluating parameters including the base quality score distribution, average base content per read, % GC distribution in the reads, and mean read length for Illumina. For the ONT reads, metrics such as the bases in GB, N50, and median read length were assessed. These evaluations were performed before commencing the hybrid WGS processing (Figure 1). Subsequently, these fastQ files underwent pre-processing using AdapterRemovalv2 [28] before performing the assembly. Initially, the process began with adapter and low-quality sequence removal, maintaining an average quality score threshold of 30 across paired-end reads. Next, duplicate reads were removed using Sequence Alignment/Map (SAMtools)’s default settings. The mapping phase yielded a total of 9.24 million reads. A sliding window of 1 kb, shifting by 100 bp at each step, was used to identify zero-coverage regions and compute the bases covered by the mapped reads (Table 1). Subsequently, de novo genome assembly was executed utilizing the Maryland Super Read Cabog Assembler (MaSuRCA) hybrid assembler v4.0.8 [29]. The obtained contigs underwent polishing using POLishing by Calling Alternatives (POLCA). Further gap filling and scaffolding was done through RagTag (https://genomebiology.biomedcentral.com/articles/10.1186/s13059-022-02823-7, 12 September 2023) with using reference *B. graminis* f. sp. *tritici* 96224 V3.16. (GCA_900519115.1). Later, the final assembled genome quality metrics -GC%, L50, L75, N50, and N75—were calculated using Quality Assessment Tool (QUAST) v4.6 [30] (Appendix A). Additionally, the average nucleotide identity (ANI) was determined using OrthoANI. The list of tools used for the genome assembly and analysis is furnished in Appendix A. Furthermore, the genome assembly was deposited into the National Center for Biotechnology Information (NCBI)/GenBank under accession number GCA_024363405.1/JALMLO000000000. We also analyzed the comparative genome assembly of *B. graminis* represented in Appendix A.

### 2.9. Molecular Phylogeny

We performed comparative genomics to establish phylogenetic relationships and calculate the average nucleotide identity (ANI) by analyzing the powdery mildew genome from this study (*B. graminis* f. sp. *tritici* Wtn1; accession number GCA_024363405.1) alongside eight other powdery mildew genomes available in the NCBI database and the documented literature (Figure 3). The fungal isolates included (i) five *B. graminis* f. sp. *tritici* isolates infecting wheat (accession numbers GCA_000418435.1, GCA_900519115.1, GCA_927323605.2, GCA_905067625.1, GCA_024363405.1); (ii) two *Erysiphe pisi* isolates infecting peas (accession numbers GCA_000208805.1, GCA_000214055.1); (iii) one *Golovinomyces cichoracearum* isolate infecting cucurbits (accession numberUMSG3GCA_003611195.1); and (iv) one *Erysiphe necator* isolate infecting grapevines (accession: GCA_016906895.1). Furthermore, the genome assembly was deposited into the NCBI/GenBank Sequence Read Archives (SRA) under the accession numberPRJNA1086707 (Figure 3).

### 2.10. Gene Prediction and Annotation

The assembled genome underwent several analyses. Repeat masking was performed using RepeatMasker v4.0.7 and Repeat Modeler v1.0.11, setting the GC content parameters to 46–48% for repeats soft masking. The AUGUSTUS v3.3.3 gene predictor was initially trained using the closest reference genome via *B. graminis* f. sp. *tritici* 96224 V3.16. Later, gene prediction was conducted using Augustus v3.3.3, which involved predicting genes, CDS and proteins independently on both DNA strands, along with the utilization of soft masking [31]. Subsequently, Benchmarking Universal Single-Copy Orthologue (BUSCO) v4.1.4 assessed the completeness of the resulting protein sequences. KofamKOALA KEGG (Kyoto Encyclopedia of Genes and Genomes) Orthology Search via HMMER KofamScanv2023-10-02, release 108.0 [32], identified the biochemical pathways, while the hierarchical functional annotations were conducted using eggNOG (evolutionary genealogy of genes: Non-supervised Orthologous Groups)-mapperv2.1.8, database v5.0.2, for further analysis [33]. The pathogen–host interaction (PHI) sequences were retrieved from the PHI database [34] using double index alignment of next-generation sequencing data (DIAMOND) v2.0.15 and a protein aligner search [35], limiting the target sequences to 1 and employing an e-value of 0.0001. The effector proteins were identified by examining the predicted total proteins for the signal peptides using the Phobius and SignalP_Euk tools available in InterProScan v5.59–91.0 [36], and subsequent analysis of the signal peptide-containing sequences was performed using Effector P v3.0, following the established methodologies [37]. Lastly, the presence of tRNA and rRNA was determined using tRNAscan-SE v0.4 and barrnap v 0.9, respectively, employing specific settings for the length threshold and e-value criteria, integrated in Galaxy platform [38].

### 2.11. Comparative Genomic Analysis

The dbCAN3 database employed the HMMER approach with an e-value of 0.00001 to identify carbohydrate-active enzymes (CAZymes) [39]. OrthoVenn3 [40] utilized the OrthoMCL algorithm to identify common and unique orthologous proteins. The gene ontology terms were retrieved using InterProScan v5.59–91.0 through a homology search across multiple databases—PFAM (protein family), Superfamily, PIRSF (Protein Information Resource family), Phobius, and SignalP_Euk—within the InterProScan v5.59-91.0 database [36]. Additionally, Blast2GO was used to classify the mapped GO IDs/terms to functional terms (PMCID: PMC2375974).

### 2.12. Transcriptomics of *B*. graminis f. sp. tritici Wtn1

*B. graminis* f. sp. *tritici* Wtn1 conidial biomass was used for the total RNA extraction and subsequent sequencing via a standard transcriptomics workflow. Enriched mRNA was used to construct a paired-end (PE) library and sequenced using the Illumina HiSeq 2500 platform, generating 4.63 GB (R1+R2) of data with a 44% GC content. Quality assessment with FastQC preceded the removal of low-quality bases and ambiguous datasets using Trimmomatic v0.35 to obtain high-quality reads [41,42]. Read mapping to the *B. graminis* f. sp. *tritici* 96224 V3.16 reference genome from the Ensembl Fungi database was performed using Bowtie2 (v2.5.0) and Cufflinks (v2.2.1.3) [43,44], considering genes with an FPKM value of ≥10 for functional annotation. Genes with fragments per kilobase of transcript per million mapped reads (FPKM) ≤ 9.0 were excluded from this annotation process. The expressed genes underwent Gene Ontology (GO) annotation via the Database for Annotation, Visualization, and Integrated Discovery (DAVID) [45,46] v2023q4. Additionally, these genes were assessed for gene enrichment using the functional annotation tool against a background reference, *B. graminis* f. sp. *hordei* DH14n [45]. Furthermore, the raw reads from the conidial transcriptome datasets are deposited to NCBI-SRA with accession number PRJNA1086707.

### 2.13. Genome Analysis for Nucleotide Variation

Initially, to determine Single-Nucleotide Polymorphisms (SNPs) and Indels, the high-quality reads were aligned with the *B. graminis* f. sp. *tritici* 96224 V3.16. (GCA_900519115.1) genome using the maximal exact match (MEM) algorithm in Burrows–Wheeler Alignment (BWA) tool v0.7.17 [47]. PCR duplicate reads from the aligned data were removed using the Mark duplicates tool within Picard (https://broadinstitute.github.io/picard/ accessed on 12 September 2023). Subsequently, variant calling was carried out employing bcftoolsmpileup from SAMtoolsv1.15.1 [48], applying filters for a minimum variant depth (DP) of ≥20, a Phred quality score (Q) of ≥25, and a mapping quality of ≥30. The identified SNPs were annotated for different genomic regions, as were their various effects and impact types using SnpEffv4.5 [49]. For SNP density plot, the variant positions were submitted to SNP density window of SR plot [50].

## 3. Results

The study investigated the virulence patterns of *B. graminis* f. sp. *tritici* isolates on diverse wheat varieties and the consequent efficacy of the *Pm* gene in conferring mildew resistance (Figure 2). To identify *B. graminis* f. sp. *tritici* Wtn1, we sequenced four pathogenesis-related genes (*PKC1* and *PKC*-like; *Cat1;* AOX; and *pth*). The sequences obtained were accessioned in the NCBI database with the accession numbers OP271713, OP087312, OP354504, and OP271716, respectively. Among the 15 assessed *B. graminis* f. sp. *tritici* isolates, *B. graminis* f. sp. *tritici* 29 (renamed *B. graminis* f. sp. *tritici* Wtn1) exhibited infectivity across various wheat varieties and near-isogenic lines like Amigo (*Pm*17), ChulBidai (*Pm*3b), and Kavakaz (*Pm*8). However, it failed to infect Timgalen, carrying the *Pm*6 gene, displaying high virulence on ChulBidai (*Pm*3b) seedlings and moderate susceptibility on the other *Pm* genes. Due to its unique virulence profile, *B. graminis* f. sp. *tritici* Wtn1 was chosen for the whole-genome analysis, corroborated by PCA analysis. Molecular phylogenetic analysis based on the ITS sequence (518-bp; GenBank Acc. No. MN872365.1) confirmed a genetic similarity of 99.6–100.0% with AB462308.1 and MN919383.1, representative of mildew isolates from different regions/countries, affirming the close genetic relationship among mildew isolates impacting wheat crops.

### 3.1. Sequence Read Quality, Genome Assembly and Evaluation

We produced 13.97 Gb of data with 92.50 million reads; over 92.7–92.0% exceeded Q30. Additionally, ONT long-read sequencing generated 4.94 Gb from 3.23 million reads. The curated raw data for the genome hybrid assembly comprised 92.41 million reads (13.34 Gb; GC-45.4%) (Table 1). Our draft-assembled genome spans 93 scaffolds with a CG content of 43.72%. It is 140.61 Mb with an N50 of 16.02 Mb. The longest contig in the hybrid assembly reached around 19.53 Mb. Genome completeness assessment revealed 95.4% complete and 1.6% fragmented BUSCOs. Comparative analysis showed a high similarity (99.26% to 99.69%) within the *B. graminis* f. sp. *tritici* genomes, contrasting with a 65% to 73% reduction in dicotyledon-host-infecting species like *E. necator*, *E. pisi*, and *G. cichoracearum*. Molecular phylogeny based on whole-genome alignment highlighted alignments among various powdery mildew fungi, showing similarities between the monocot-infecting and dicot-infecting species (Figure 3). Furthermore, BLASTP analysis indicated significant matches with *B. graminis* f. sp. *triticale*, *B. graminis* f. sp. *tritici*, *B. graminis* f. sp. *tritici* 96224, and *B. hordei*. The organism chart revealed the highest hit for the test isolate, followed by *B. graminis* f. sp. *hordei* and *B. graminis* f. sp. *hordei* strain DH14 (Appendix A).

### 3.2. Identification of Repeats and Transposable Elements

The assembled genome showed repetitive sequences, encompassing 73.8% of its entirety (Table 2). Among these, retro-elements predominated, with 95,626 copies covering 62.24% of the genome, notably LINEs accounting for 42.92 Mb. LTR elements were the second most abundant, featuring 59,884 copies across 43.122 Mb (31.16% of the genome). Gypsy/DIRS1 and Copia elements were prominent within this category, comprising 15.29% and 14.43% of the genome, respectively. DNA transposons and Tc1-IS630-Pogo elements had almost equal proportions, each occupying 1.13 Mb and constituting 0.82% of the genome. Interspersed repeats represented 73.80% of the sequences, spanning a length of 102 Mb (102,143,698) (Table 2).

### 3.3. Gene Prediction, Annotation Validation, and Comparative Protein Coding Genes

InterProScan Analysis. The gene and protein prediction analysis revealed 8480 protein-encoding genes (PEGs) and 190 rRNA and 583 tRNA sequences (Figure 4). Within the Wtn1 isolate, the protein-coding density (PCD) covered a total length of 11,135,169 coding bases (CDS) occupying a 12,710,518 gene length; encoded 3,704,012 protein amino acids, averaging a length of 1649 genes; and had a 1445 CDS length translating to the 481 protein-encoding genes (PEGs). Determination of the gene density found it to be unevenly distributed in each chromosome (Chr). We also observed that the Chr 1, 2, 7, 8, and 10 end positions/regions had a high gene density compared to the other chromosomes. Further, Chr1 to 10 have a medium gene density, whereas the Chr 3 terminal region has the lowest gene density (Appendix A). InterProScan analysis unveiled diverse functional features, identifying 3180 distinct protein families (PFAMs), detecting 7845 PFAMs in the genome (Figure 5). Additionally, the analysis recognized 774 distinct superfamilies with 5699 copies, 357 PIRSF with 468 copies, and 2179 FunFam features comprising 3218 copies within the *B. graminis* f. sp. *tritici* Wtn1 genome. Comparative proteome analysis showed a substantial overlap, with 86%, 90%, 71%, and 76% of the protein families, superfamilies, PIRSF, and FunFam features, respectively, shared among the three studied genomes (Figure 5). Notably, there was a significant sharing of PFAMs, superfamilies, and FunFam features between the *B. graminis* f. sp. *tritici* Wtn1 and *B. graminis* f. sp. *triticale* THUN-12 genomes. The expanded PFAMs in these genomes included fungal protein kinases; the WD domain, G-beta repeat; the protein kinase domain; RNA recognition motifs (RRM, RBD, and RNP domain); mitochondrial carrier proteins; and helicase conserved C-terminal domains.

### 3.4. Analysis of the Orthologous Genes and CAZymes in the Assembled Genomes

Exploring the orthologous genes among *B. graminis* f. sp. *tritici* Wtn1 and other genomes found 5139 common proteins across the studied genomes. Notably, distinct orthologous genes were present in *B. graminis* f. sp. *tritici* Wtn1 (111 genes), *B. graminis* f. sp. *tritici* 96224 v3.16 (134 genes), and *B. graminis* f. sp. *triticale* THUN-12 (61 genes) (Figure 6). Regarding CAZymes, the genome displayed 156 CAZyme families. The most abundant were glycoside hydrolases (70-GH), glycosyltransferases (58-GT), and auxiliary activities (13-AA) (Table 3). The other identified families included carbohydrate esterase (11-CE) and carbohydrate-binding modules (4-CBM). Surprisingly, pectin lyase (PL) was absent in all the compared genomes. The comparative analysis revealed that 73 (90%) CAZyme families were shared among different *B. graminis* forma specialis. Notably, distinctive observations included *B. graminis* f. sp. *tritici* Wtn1 having two GH subfamily CAZymes, GH31_1 and GH55_2, while THUN-12 possessed a unique GH31 (Figure 7).

### 3.5. Deciphering the tRNA Anticodons in the Genome

Within the *B. graminis* f. sp. *tritici* Wtn1 genome, 583 tRNA anticodon sequences were identified, including the stop codon. The most prevalent anticodons were for leucine (70), cysteine (61), and alanine (58), followed by arginine (45), serine (37), and methionine (31). Aromatic amino acids were also detected, with tyrosine (21) being the most abundant and then phenylalanine (19) and tryptophan (11). Notably, five anticodons were complementary to the stop codon (Figure 8A). Examination of each anticodon triplet encoding amino acids showed varying frequencies, with GCA (Cys), TGC (Val), TAG (Leu), CAT (Met), and GTA (Tyr) being the most prevalent (Figure 8B). Notably, two anticodons, TCA and TTA, complementary to the stop codons TGA (Opal) and TAA (Ochre), respectively, were detected. Interestingly, the stop codon TAG (Amber) was absent in the *B. graminis* f. sp. *tritici* Wtn1 genome, indicating the presence of all 20 standard amino-acid-encoding codons.

### 3.6. Gene Ontology Annotation

In the Gene Ontology (GO) mapping and annotation, the genes were diversified across biological processes, molecular functions, and cellular components (Appendix A). At level 2 of the biological processes, the gene classifications ranged from 13 to 2856, while the molecular function annotations varied from 5 to 2316. For the cellular components, the gene classifications spanned from 1013 to 3004 terms. Among the biological processes, cellular processes (~2865 genes) were prominent, followed by metabolic processes (~2368 genes), localization (663 genes), and response to stimuli (384 genes). In molecular functions, “catalytic activity” had the highest representation, involving around 2316 genes, followed by binding (2204 genes), ATP-dependent activity (261 genes), transporter activity (241 genes), and molecular function regulator activity (128 genes). Considering the cellular components, genes associated with cellular anatomy activities were prevalent, followed by intracellular components and protein-containing complexes in the *B. graminis* f. sp. *tritici* Wtn1 genome.

### 3.7. KOG Classification of the Genome

Using euk-KOG classification, the genome was categorized into various functional groups. The prominent categories comprised carbohydrate transport and metabolism (G); replication, recombination, and repair (L); post-translational modification, protein turnover, and chaperones (O); intracellular trafficking, secretion, and vesicular transport (U); RNA processing and modification (A); and translation, ribosomal structure, and biogenesis (J). These prevalent features signify vital roles in the organism’s biological processes and metabolic dynamics (Figure 9). Additionally, the overall functional categories encompassed transcription (K), amino acid transport and metabolism (E), lipid transport and metabolism (I), and the cytoskeleton (Z), indicating their involvement in fundamental cellular mechanisms and developmental processes.

### 3.8. KEGG Pathways

In the *B. graminis* f. sp. *tritici* Wtn1 genome, diverse functional processes were observed in the analysis of the biochemical pathways. Under carbohydrate metabolism (KO:09101) within the broader metabolic pathways (KO:09100), 213 enzymes were identified across fifteen pathways, including Pyruvate metabolism, Glycolysis/Gluconeogenesis, starch and sucrose metabolism, Inositol phosphate metabolism, and Glyoxylate and dicarboxylate metabolism (Table 4). Additionally, the investigation highlighted 16 genes/enzymes involved in the Terpenoid backbone biosynthesis pathway within the metabolism of terpenoids and polyketides (Table 5). In the Environmental Information Processing pathways, 393 genes/enzymes were found in 33 signal transduction pathways, notably in the MAPK signaling pathway–yeast (53 genes), the mTOR signaling pathway (38 genes), PI3K-Akt signaling (23 genes), and AMPK signaling (21 genes) (Table 6).

### 3.9. KEGG Mapper BRITE Classification

In analyzing the genes/enzymes associated with ko00001 in KEGG Orthology (KO), 2804 genes were identified and categorized into three prominent protein families: metabolism, genetic information processing, and signaling. In the metabolism-related protein family, diverse enzymes like protein kinases (58), peptidases and inhibitors (89), glycosyltransferases (37), and amino-acid-related enzymes (32) were found (Appendix A). The protein kinase analysis revealed distinct kinase groups, including serine/threonine kinases (AGC, CMGC, and STE), Histidine kinases, and prominent families like CAMKL, CDK, MAPK, DYRK, STE3, PIKK, and RIO. Among the peptidases and inhibitors, various categories like Aspartic, cysteine, metallo, serine, and threonine peptidases were identified, with families like the C19: ubiquitin-specific protease, M16: pitrilysin, M28: aminopeptidase Y, M41: FtsH endopeptidase, S10: carboxypeptidase Y, S26: signal peptidase I, and T1: proteasome families notably present in the Wtn1 genome. In the genetic-information-processing-related protein family, the prevalence of zinc finger transcription factors (TFs) was observed, alongside other TFs such as helix–turn–helix, basic leucine zipper (bZIP), AP-1(-like) components, fungal regulators, Cys2His2, MADS-box, and heteromeric CCAAT factors. The chaperone and folding catalyst analysis demonstrated a high abundance of heat shock proteins (HSP20, HSP40/DNAJ, HSP60/chaperonin, GimC, subtilisin family) and protein-folding catalysts like cyclophilin and protein disulfide isomerase in the *B. graminis* f. sp. *tritici* Wtn1 genome.

### 3.10. Prediction of Pathogenicity and Virulence Effectors

The genome analysis identified 805 predicted genes encoding effector proteins, with 511 linked to pathogenesis (Appendix A). Notably, eleven common pathogenesis-related genes were found in the *B. graminis* f. sp. *tritici* Wtn1 genome, including *ACE1*, *AGLIP1*, *Avra10*, *Avrk1*-like, *BcCLA4*, *BEC1005*, *BEC1016*, *BEC1040*, *B. graminis* f. sp. *tritici*_Bcg-7, *CSEP0105*, *CSEP0162*, *HopI1*, and *MoCDIP4*. Using the pathogen–host interaction database (PHI category), 279 genes showed reduced virulence, 105 were unaffected, 39 lost pathogenicity, and 36 were lethal. Further, a total of 2104 protein sequences with signal peptide features were processed using SignalP_EK and Phobius, and these were submitted to effectorP3.0, which employs a machine learning approach). This analysis identified 805 predicted effectors in the *B. graminis* f. sp. *tritici* Wtn1 genome. Among these, 720 (34.2%) were cytoplasmic, and 85 (4.0%) were apoplastic. Additionally, 10.1% of the cytoplasmic effectors were expected to have dual localization (cytoplasmic/apoplastic), while 23.5% of the apoplastic effectors were predicted to have dual localization (apoplastic/cytoplasmic).

### 3.11. Transcriptome-Driven Gene Expression and Functional Annotation

Transcriptome mapping of the conidia data demonstrated that 84.62% was exonic, 7.74% was intronic, and 7.74% was intergenic (Appendix A). In the strain Wtn1 conidia, 4234 genes showed expression levels with FPKM values ≥ 10. These genes were classified based on their FPKM values: very high (≥1000 FPKM), high (≥500–999 FPKM), medium (≥100–499 FPKM), and low (≥10–99) expression levels. Notably, there were 43 (1.02%) genes identified as very highly expressed, 70 (1.65%) as highly expressed, 815 (19.25%) as moderately expressed, and 3306 (78.08%) as lowly expressed genes within *B. graminis.*
Appendix A contains the details of the 30 genes categorized as highly expressed (≥1000 FPKM) in the conidia.

### 3.12. Functional Annotation of the Conidial Transcriptome

Transcript profiling analysis of the conidia revealed genes associated with biological processes, molecular functions, and cellular components (*p*-value < 0.01). Under biological processes, 37 enriched terms included translation (96 genes), glutamine metabolic process (14 genes), proteasome-mediated ubiquitin-dependent protein catabolic processes (12 genes), biosynthetic processes (10 genes), carbohydrate metabolic processes (36 genes), protein folding (22 genes), and lipid catabolic processes (9 genes). Among 38 enriched terms under cellular components were the ribosome (86 genes), mitochondrial inner membrane (57 genes), large ribosomal subunit (11 genes), chromatin (11 genes), and cytoplasm (148 genes). The molecular function analysis revealed 46 enriched terms, including ATP binding (270 genes), structural constituents of the ribosome (107 genes), mRNA binding (14 genes), transferase activity (33 genes), metal ion binding (157 genes), oxidoreductase activity (41 genes), hydrogen ion transmembrane transporter activity (9 genes), peroxidase activity (9 genes), calcium ion binding (22 genes), translation initiation factor activity (38 genes), and GTPase activity (40 genes). Analysis of the expressed genes revealed significant enrichment in various protein families, including the WD domain, G-beta repeat (50); RNA recognition motif (40); DEAD/DEAH-box helicase (26);proteasome subunit (13), ubiquitin-conjugating enzyme (13), ATPase associated with various cellular activities (AAA) (21); and cytochrome b5-like heme/steroid-binding domain (9) PFAMs. Thirty highly expressed genes in the conidia are outlined (Appendix A).

### 3.13. SNPs and Indels in *B*. graminis f. sp. tritici

The SNP and Indel analysis revealed 356,076 variants within this genome, including 345,375 SNPs, 5050 insertions, and 5651 deletions. When categorized by chromosome, the highest SNP count was observed on chromosome 5, followed by chromosomes 9, 7, and 2 (Figure 10 and Figure 11a). The SNP density was found asymmetric or unevenly distributed in each chromosome. We also observed that the Chr 5, 6, 7, 9 and 10 start and end positions/co-ordinates have a high SNP density compared to the other Chrs. Further, Chrs 1, 2, 4, and 11 have a moderate SNP density (Figure 10). The average SNP density per chromosome ranges from 2782 to 49,913 (Figure 10). The transition (Ts) and transversion (Tv) counts were determined to be 377,264 and 232,101, respectively, resulting in a Ts/Tv ratio of 1.63. The annotation of these variants involved classification based on region, impact, and functional class (Figure 11). Region-wise classification of the identified variants indicated that the majority were Intergenic (43.17%), followed by the downstream (26.69%) and upstream (26.46%) regions (Figure 11b). Functional class analysis revealed that 52.03% were missense variants, and 46.87% were silent (Figure 11c). Impact-wise classification showcased that the majority of variants were labeled as modifiers (98.6%), with fewer categorized as having a moderate or low impact (Figure 11d).

## 4. Discussion

Globally, wheat cultivation is threatened not only by rusts and blast but also by the powdery mildew caused by *B. graminis* f. sp. *tritici* [24,51], disseminated by airborne conidia [19]. Mildew management relies heavily on the application of fungicides and the cultivation of resistant wheat varieties [52,53]. However, the emergence of new pathotypes and races renders these strategies ineffective and short-lived. Recent genomic technologies and approaches offer new opportunities for durable crop disease management in agriculture [54,55,56]. In India, powdery mildew in wheat is an emerging disease, especially in hilly terrains. Despite its importance, and economic significance, so far, no attempts have been made to generate genomic data and genome resources for the Indian strain of *B. graminis* f. sp. *tritici*. Challenges such as high conidial moisture and difficulties in obtaining contamination-free high-quality genomic DNA hinder comprehensive genome analysis. In the past, conidial biomass has been used for genomic DNA isolation from related mildew strains affecting barley, wild grass, rye, and Lolium [57]. Environmental DNA samples are expected to carry unintended microorganisms and host plant DNA [58].

Initially, we ensured the genetic purity of *B. graminis* f. sp. *tritici* through in vitro culturing on detached and surface-disinfected wheat leaves, following the established methods [17,59,60]. Molecular–taxonomic analysis using ITS sequencing and four other gene sequences confirmed its identity as *B. graminis* f. sp. *tritici* [61] (Ac. No. MN872365.1). We used fungal biomass generated from monoconidial cultures for the genome sequencing following several methods [55]. Employing strategies to minimize DNA degradation, protein and carbohydrate contamination, and deaminating the nucleotides [24,62,63,64,65,66], we successfully isolated high-molecular-weight genomic DNA. Notably, Sarcosyl proved effective due to its ability to dissociate the nucleosomes and ribosomes while inhibiting nucleases. Utilizing a hybrid approach, we assembled short and long reads, resulting in 93 contigs totaling 140.61 Mb with a GC content of 43.7%. In recent times, integrating genome sequencing advancements and hybrid assembly algorithms has significantly accelerated genome analysis, functional predictions, and characterization [67].

The nucleotide sequence hits aligned with *B. graminis* f. sp. *tritici* identified well-assembled genome and annotated genes [68]. The use of the average nucleotide identity(ANI) revealed two distinct *Blumeria* clades, revealing genetic variability without differences in virulence across various hosts [17]. The genome showed 95,626 retro-elements, covering 86.14 Mb (62.24%), indicating a proliferation of transposable elements (TEs) contributing to virulence and ecological and host adaptations [18,69]. TEs contribute to virulence evolution by introducing genetic variations through insertions and strand invasions [70]. In particular, their mobility within genomes leads to gene duplication, horizontal gene transfer, and gene loss, influencing the evolution of virulence factors [13]. In powdery mildew, the TE proliferation results in the absence of the RIP pathway, a rarity among related ascomycetes [18]. Around 74% of the *B. graminis* f. sp. *tritici* genome consists of TE-derived sequences distributed across the chromosome arms alongside genes [22]. Pathogens with larger genomes often harbor increased repetitive DNA, particularly retrotransposons [71,72], aligning with the extensive TE coverage observed in the *B. graminis* f. sp. *tritici* Wtn1 genome. Previous studies have linked larger genomes to a higher abundance of repeated elements [73,74].

The annotated *B. graminis* f. sp. *tritici* genome is predicted to possess ~8000 protein-coding genes, which is relatively low for filamentous fungi, potentially due to the conversion to pseudogenes and the prevalence of repeat elements. This genome lacks the Amber stop codon, containing only the Opal and Ochre tRNA anticodons. Previous studies have identified 6540 genes, with 5258 shared between *B. graminis* f. sp. *tritici* and *B graminis* f. sp. *hordei* [19,75]. The Indian isolate *B. graminis*Wtn1 exhibited higher gene content than these previous findings, with 8480 genes. Fungal pathogens leverage diverse effectors to expand their host range fitness [76]. Analysis of 36 *B. graminis* f. sp. *tritici* isolates suggests that effector genes are vulnerable to duplication and deletion, possibly due to the high presence of repeat elements [22]. Our data revealed 805 predicted effector genes in the *B. graminis* f. sp. *tritici* genome, including established ones like *B. graminis* f. sp. *tritici*_Bcg-7, *BEC1005*, *CSEP0162*, *BEC1016*, and *BEC1040*. Several of these effectors are reported to enter into wheat cells via specialized feeding structures called haustoria [77].

Our study further detected 8 Avr genes among detected pathogenesis-related genes in the *B. graminis* f. sp. *tritici* Wtn1 genome, aligning with prior findings [78,79]. The Avr genes identified in earlier studies within *B. graminis* might serve as effectors, potentially contributing to its aggressiveness [79,80,81]. Understanding these genes can offer insights into the functions of R-genes, akin to *AVRa10* and *AVRk1* in barley, influencing infection in susceptible varieties and potentially sharing similar functions in *Triticum* species [82]. Genomic effector identification is important for unravelling host–pathogen interactions [83,84]. Virulence genes can provide insights into how wheat genotypes respond to powdery mildew infections, aiding the development of resilient wheat cultivars. Powdery mildews like *B. graminis* f. sp. *tritici*, *Erysiphe pisi*, and *Golovinomyces cichoracearum* have undergone specific gene loss, suggesting host-specific adaptations [18]. The mildews encode diverse secreted proteins, potentially vital to species-specific adaptation, associated with biological processes like secondary metabolite biosynthesis, transport, signal transduction, and DNA functions, contributing to their adaptability. Understanding these interactions is vital to breeding resilient wheat varieties against powdery mildew epidemics.

RNA sequencing of the conidia of *B. graminis* f. sp. *tritici* Wtn1 uncovered 30 highly expressed genes among the 4234 expressed genes. Notably, genes related to carbon and nitrogen metabolism, such as glucose-repressible protein, Oleate-induced peroxisomal protein POX8, Superoxide dismutase (SOD), Glycosidase, F-box domain-containing protein, and Catalase-peroxidase, exhibited substantial expression levels [18,20,21,22,85,86]. Regarding the Carbohydrate-Active Enzyme (CAZyme) families, the glycoside hydrolase (GH) families predominated, followed by the glycosyl transferase (GT) and auxiliary activity (AA) families in the *B. graminis* f. sp. *tritici* Wtn1 genome. These enzymes are essential in breaking down plant cell walls, aiding pathogen entry and colonization, particularly in degrading the complex polysaccharides found abundantly in monocots. The genome of *B. graminis* f. sp. *tritici* Wtn1 also revealed peptidase families such as M28, carboxypeptidase Y, subtilisin, and signal peptidase I, important to various physiological processes and consequent fungal nutrition, development, and pathogenesis [87,88]. Additionally, chaperones, folding catalysts, and protein kinases suggest their involvement in stress adaptation, temperature resilience, and enzymatic regulation through phosphorylation for activation or deactivation [89]. These diverse enzyme families and regulatory elements significantly contribute to fungal adaptation, pathogenicity, and the intricate interactions between the pathogen and its host [85,86,88].

InterProScan analysis of both the genome and transcriptome revealed an expansion of diverse protein families, showcasing functional diversity critical for the fungus to adapt within the host environment [69]. The genome also highlighted transport families like ABC transporters, sugar transporters, and Major Facilitator Superfamily (MFS) proteins, which are essential for nutrient and supplement movement within the organism. The ABC transporter R-genes in *Fusarium graminearum* are associated with azole tolerance and defense against environmental toxic compounds [90]. In barley powdery mildew, the MAPK and cAMP signaling pathways regulate various crucial aspects, including appressorium development, growth, disease progression, chemotaxis, virulence, and secondary metabolite production [91]. Similarly, in *B. graminis* f. sp. *tritici*, diverse biochemical pathways and orthologous functional features contribute to essential metabolites and intermediate compounds necessary for cellular development and infection proliferation.

Multiple studies emphasize the significance of signaling pathways like MAP kinase and cAMP in pathogen growth and disease development [90,92]. Combined with the two-component system, these pathways play a key role in disease progression and the formation of infectious structures within the host [93]. cAMP signaling influences fungal development and infection, while Ras signaling governs pathogenicity and morphogenesis [93,94,95,96]. These pathways regulate chemotaxis, phosphorylation-mediated signaling, virulence, and secondary metabolite production [97]. Protein domains forming multidomain assemblies contribute significantly to functional diversity [98,99], crucial to the pathogen’s virulence, adaptation to the host environment, and disease-causing abilities [100].

The abundance and distribution of variants, particularly the higher occurrence of SNPs on specific chromosomes like 5, 9, and 2 within the *Blumeria* genome, signify a mosaic of genetic diversity and potentially distinct evolutionary pressures. The observed bias for transitions over transversions (Ts/Tv ratio of 1.63) suggests a preference for specific nucleotide changes, possibly reflecting mutational patterns or selective constraints in the evolutionary trajectory of *Blumeria*. The prevalence of intergenic variants implies potential regulatory alterations that might influence gene expression, adaptation, or responses to selective forces. The dominance of missense variants indicates potential changes in the protein-coding sequences, possibly contributing to pathotype evolution in *Blumeria*. Moreover, the predominance of modifier impact variants suggests nuanced alterations in molecular functions, possibly contributing to the fine-tuning of evolutionary strategies related to wheat host interactions.

## 5. Conclusions

The genome and transcriptome data of *B. graminis* f. sp. *tritici* uncover critical insights into its diverse metabolic pathways, pathogenicity & effector genes, and virulence mechanisms, showcasing potential threats to wheat cultivation. Retro-elements, LTR elements, P elements, diverse protein families, and signaling factors, coupled with abundant tRNA anticodons, likely drive its evolution. Phylogenomic analysis places this isolate in a distinct monocot-infecting clade. Its genetic makeup with SNP clusters, a transition bias, and diverse variants, including impactful missense and modifier variants, reflects its adaptive evolution, influenced by unique pressures and wheat interactions. The abundance of proteins related to RNA processing, post-translational modification, protein turnover, chaperones, and signal transduction pathways presents opportunities for developing effective, long-lasting strategies for managing the powdery mildew pathogen and its threats to the production and quality of wheat in India.

## Figures and Tables

**Figure 1 jof-10-00267-f001:**
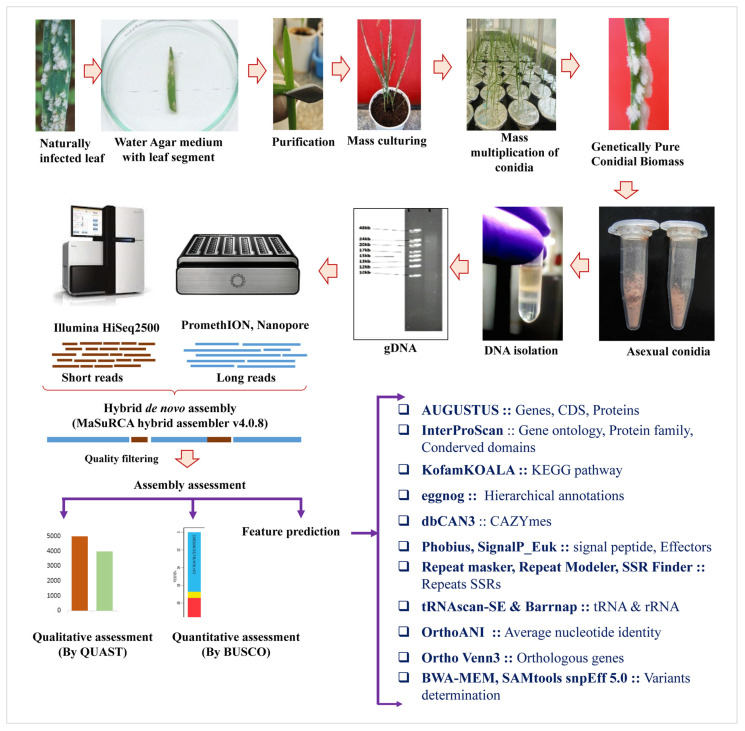
Schematic illustration of WGS analysis of *B. graminis* f. sp. *tritici* Wtn1.

**Figure 2 jof-10-00267-f002:**
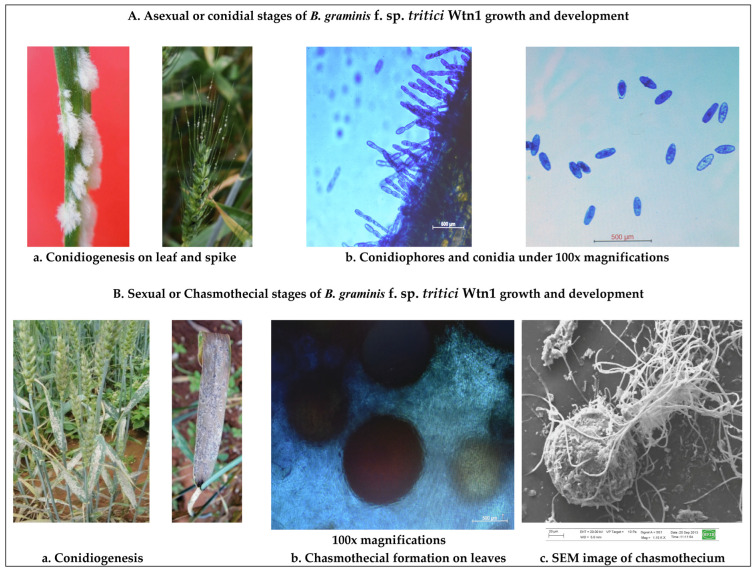
(**A**) (**a**,**b**) depict the symptoms and mycological characteristics of wheat powdery mildew pathogen *B. graminis* f. sp. *tritici* Wtn1 at hot-spot location (Wellington) in Nilgiris hills of India. (**B**) (**a**). Conidiogenesis with airborne conidia of wheat powdery mildew pathogen expressed on leaves, stem, and awns of susceptible cultivar WL 711. (**b**). Formation of black and round-shaped chasmothecia on leaf surface. (**c**) Chasmothecium with paraphyses under SEM analysis.

**Figure 3 jof-10-00267-f003:**
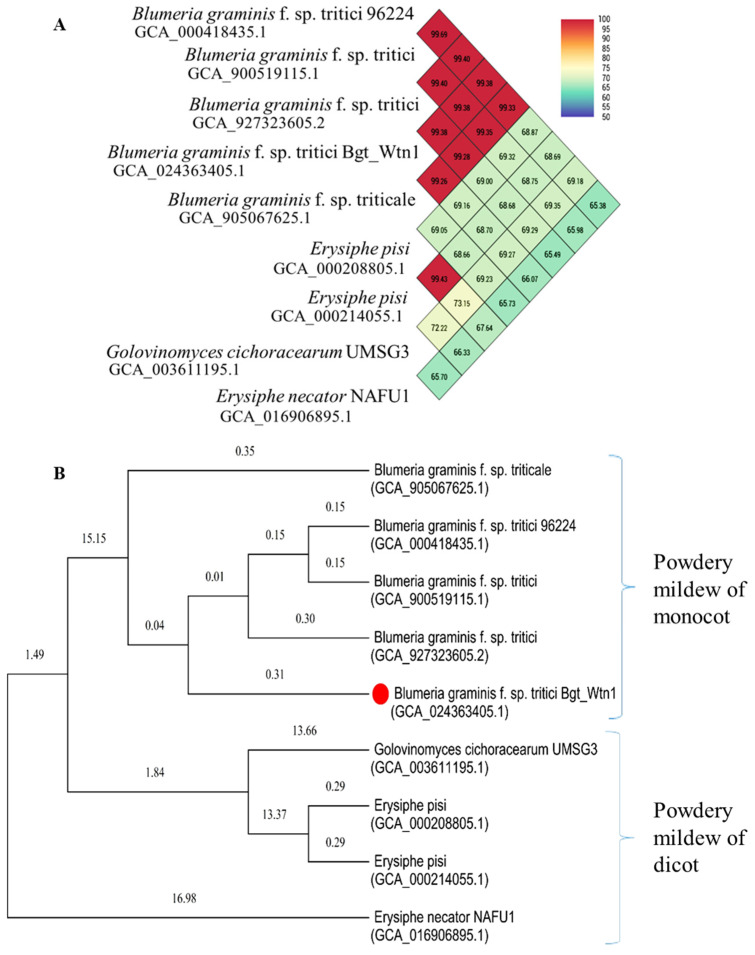
(**A**)Average nucleotide identity (ANI) based on WGS alignment of powdery mildew fungal genera of different crop plants; (**B**) phylogenetic tree constructed based on WGS alignment of powdery mildew fungal genera of different crop plants.

**Figure 4 jof-10-00267-f004:**
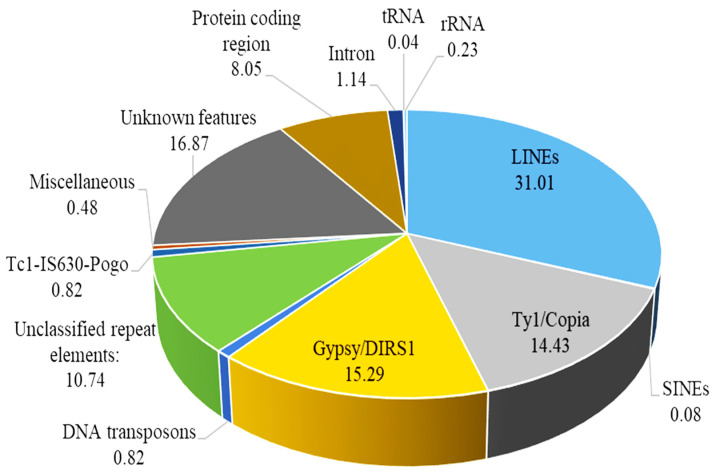
Overview of different components that make up the *B. graminis* f. sp. *tritici* Wtn1 genome. Majority of the genome composed of non-coding sequences, such as introns and transposable elements, including long interspersed nuclear elements (LINEs), short interspersed nuclear elements (SINEs), and long terminal repeat (LTR) retrotransposons. Only 8.05% of the genome actually consists of protein-coding regions.

**Figure 5 jof-10-00267-f005:**
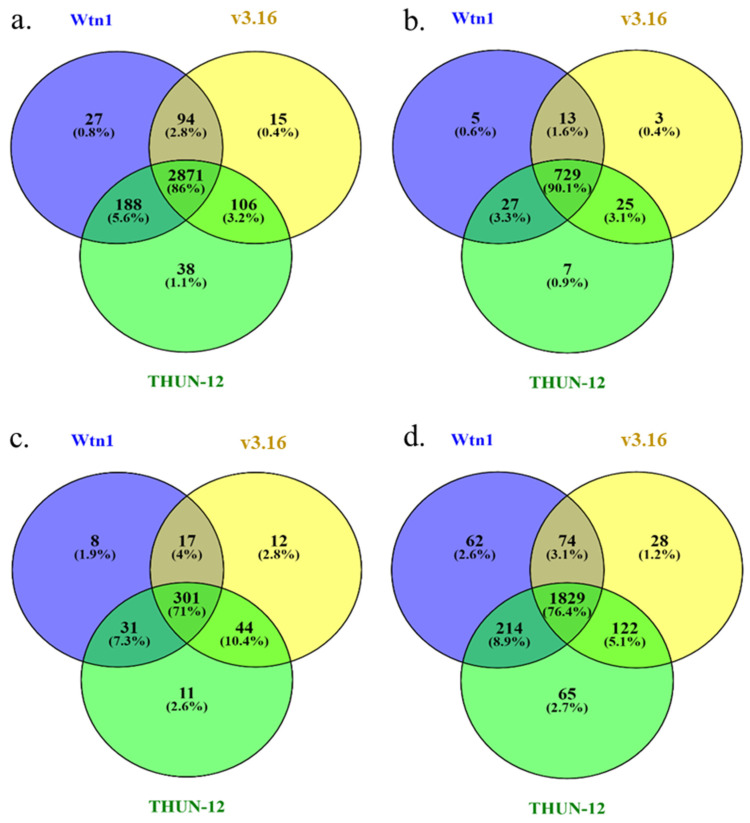
The comparative InterProScan analysis of *B. graminis* f. sp. *tritici* Wtn1, *B. graminis* f. sp. *tritici* 96224 V3.16, and *B. graminis* f. sp. *triticale* THUN-12 genome predicted proteins. The figures (**a**–**d**) show the analyses against PFAM, superfamily, PIRSF and FunFam databases, respectively.

**Figure 6 jof-10-00267-f006:**
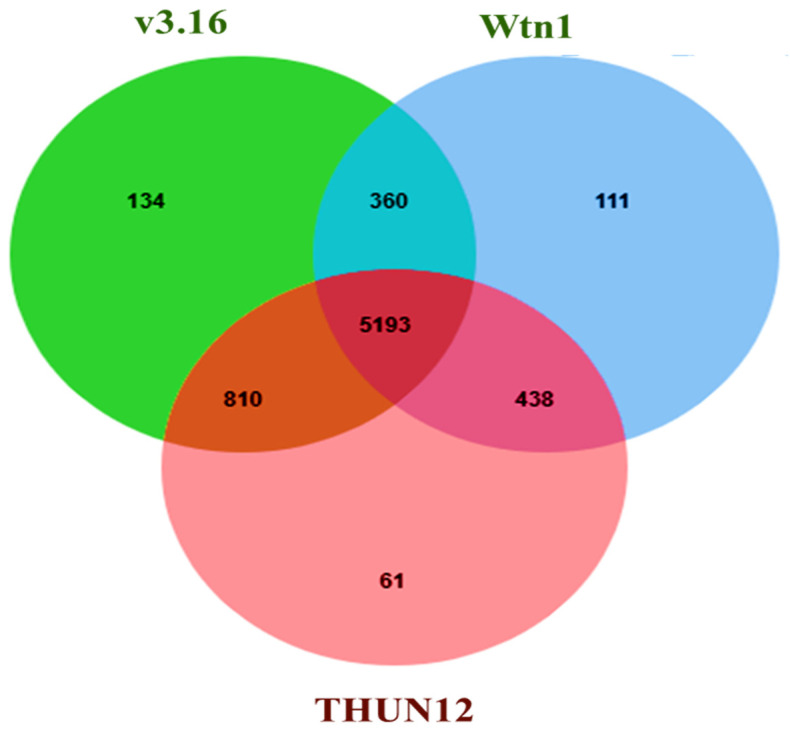
Venn plot showing orthologous proteins among the *B. graminis* f. sp. *tritici* Wtn1, *B. graminis* f. sp. *tritici* 96224 V3.16, and *B. graminis* f. sp. *triticale* THUN-12 reference genomes.

**Figure 7 jof-10-00267-f007:**
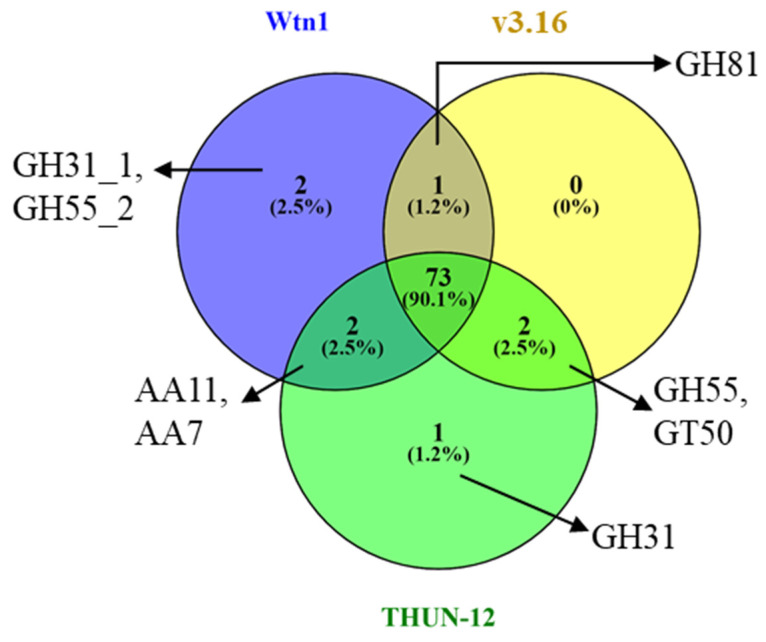
Identification and comparison of unique and shared CAZymes in *B. graminis* f. sp. *tritici* Wtn1, *B. graminis* f. sp. *tritici* 96224 V3.16, and *B. graminis* f. sp. *triticale* THUN-12 genomes. The common CAZyme families in all compared/studied genomes are given here: AA1_2, AA1_3, AA2, AA3_2, AA4, AA5_1, AA9, CBM20, CBM21, CBM43, CBM87, CE1, CE16, CE18, CE3, CE4, CE5, GH114, GH125, GH128, GH13_25, GH13_40, GH13_8, GH131, GH132, GH135, GH15, GH152, GH16_1, GH16_18, GH16_19, GH16_2, GH16_22, GH16_3, GH17, GH18, GH20, GH3, GH37, GH38, GH47, GH5_12, GH5_9, GH63, GH72, GH76, GH78, GH92, GH93, GT1, GT15, GT2, GT20, GT21, GT22, GT24, GT3, GT32, GT33, GT34, GT35, GT39, GT4, GT48, GT57, GT58, GT59, GT62, GT66, GT69, GT76, GT8, and GT90.

**Figure 8 jof-10-00267-f008:**
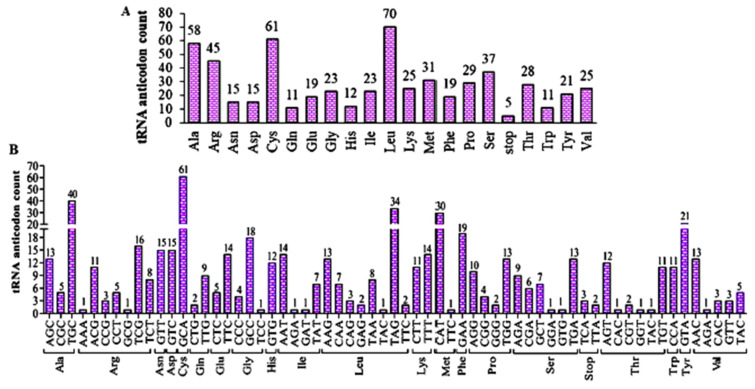
Identified tRNA anticodon sequences in *B. graminis* f. sp. *tritici* genome. (**A**) Identified total anticodons corresponding to amino acids. (**B**) Each anticodon corresponding to an amino acid. The abbreviated three letters indicate anticodons corresponding to coding amino acids. The abbreviated amino acids stand for Ala = Alanine, Arg = Arginine, Asn = Asparagine, Asp = Aspartic Acid, Cys = Cysteine, Glu = Glutamic Acid, Gln = Glutamine, Gly = Glycine, His = Histidine, Ile = Isoleucine, Leu = Leucine, Lys = Lysine, Met = Methionine, Phe = Phenylalanine, Pro = Proline, Ser = Serine, Thr = Threonine, Trp = Tryptophan, Tyr = Tyrosine, Val = Valine, Stop = Non-Coding.

**Figure 9 jof-10-00267-f009:**
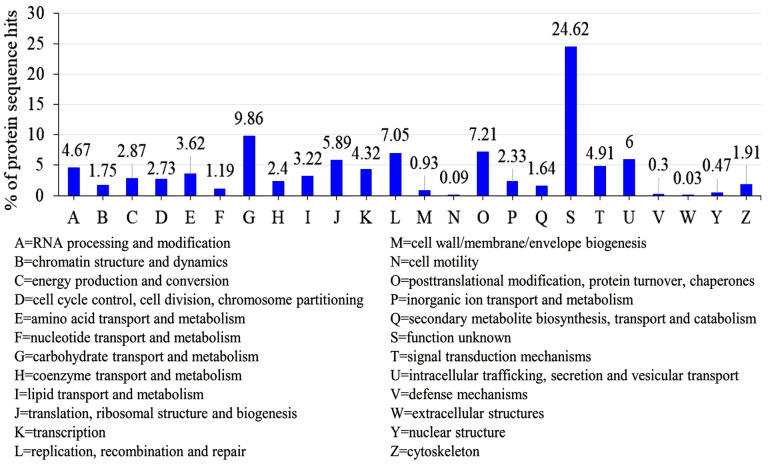
Eukaryotic orthologous groups of proteins in *B. graminis* f. sp. *tritici* Wtn1 genome. eggNOG was used for functional feature determination.

**Figure 10 jof-10-00267-f010:**
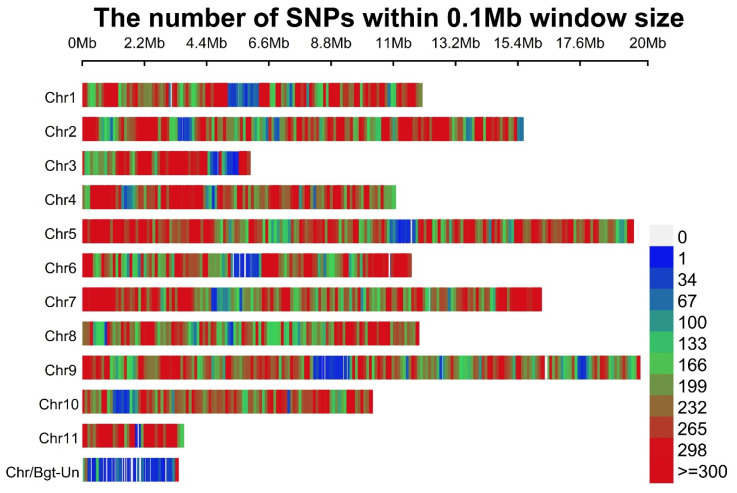
The identified SNP density plot with respect to chromosomes representing number of SNPs within 0.1 Mb window size in *B. graminis* f. sp. *tritici* Wtn1 genome.

**Figure 11 jof-10-00267-f011:**
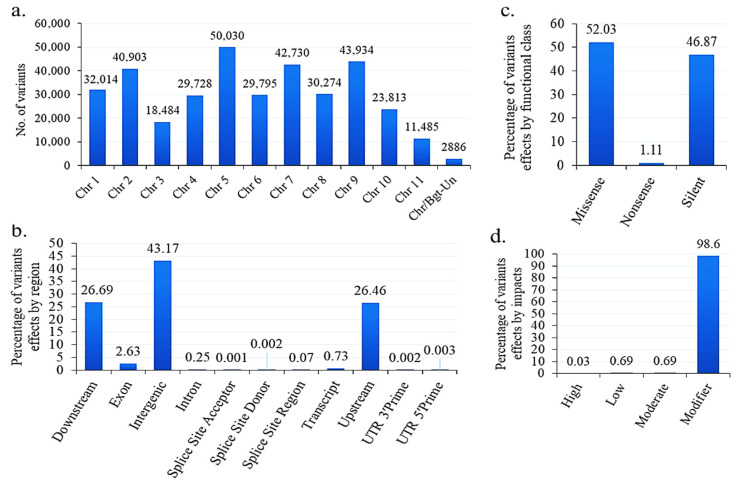
Identified variants and effect annotation *B. graminis* f. sp. *tritici* Wtn1 genome: (**a**) variant distribution by chromosome, (**b**) effect by different region, (**c**) effect by functional class, and (**d**) effect by impact.

**Table 1 jof-10-00267-t001:** Statistics of the assembled whole genome of *B. graminis* f. sp. *tritici* Wtn1.

Attributes	*B. graminis* f. sp. *tritici* Wtn1
Illumina HiSeq 2500 Raw (PE)	92,496,414 (PE reads), 13.97 Gb
HiSeq 2500 Clean (PE)	9,24,13,253 (PE reads), 13.34 Gb
Illumina Nanopore PromethION	32,25,945 (Single end), 4.94 Gb
Scaffolds	93
Largest contig	19,534,821
Total length	140,604,965
GC (%)	43.72
N50	16,029,079
N75	9,486,802
L50	47
L75	101
No. of genes	8480
Proteins	7707
rRNA	190
18S	61
28S	61
5S	6
5.8S	62
tRNA	583
CAZymes	156
Repeat element bases	102.14 Mb (73.8%)
#N’sper100 kbp	0.07
Complete BUSCOs (C)	723 (95.4%)
Complete and single-copy BUSCOs (S)	714 (94.2%)
Complete and duplicated BUSCOs (D)	9 (1.2%)
Fragmented BUSCOs (F)	12 (1.6%)
Missing BUSCOs (M)	23 (3.0%)
Total BUSCOs	758 (100%)

**Table 2 jof-10-00267-t002:** Statistics of the repeat elements in assembled whole genome of *B. graminis* f. sp. *tritici* Wtn1.

Repeat Elements	Number of Elements	Length Occupied (bp)	Percentage of Sequence
**Retro-elements**	95,626	86,147,131	62.24
SINEs:	1118	106,274	0.08
Penelope	7	317	0.00
LINEs	34,624	42,918,846	31.01
CRE/SLACS	2	398	0.00
L2/CR1/Rex	0	0	0.00
R1/LOA/Jockey	9	685	0.00
R2/R4/NeSL	0	0	0.00
RTE/Bov-B	0	0	0.00
L1/CIN4	0	0	0.00
**LTR elements:**	59,884	43,122,011	31.16
BEL/Pao	0	0	0.00
Ty1/Copia	26,614	19,968,861	14.43
Gypsy/DIRS1	26,883	21,158,915	15.29
Retroviral	0	0	0.00
DNA transposons	2358	1,136,913	0.82
hobo-Activator	0	0	0.00
Tc1-IS630-Pogo	2334	1,135,200	0.82
En-Spm	0	0	0.00
MuDR-IS905	0	0	0.00
PiggyBac	0	0	0.00
Tourist/Harbinger	7	414	0.00
Other (Mirage)	2	51	0.00
**P-element, transit**			
Rolling-circle	0	0	0.00
Unclassified	29,224	14,859,654 bp	10.74
Small RNA	150	305,896	0.22
Satellites	284	70,281	0.05
Simple repeats	1183	27,3126	0.20
Low complexity	80	12,044	0.01
Total interspersed repeats: 102,143,698 bp			73.80

**Table 3 jof-10-00267-t003:** Comparative analysis of CAZyme family profile in *B. graminis*.

Category	*B graminis* f. sp. *tritici* 96224 v3.16	*B graminis* f. sp. *triticale* THUN-12	*B graminis* f. sp. *tritici* Wtn1
AA	9	13	13
CBM	4	4	4
CE	11	10	11
GH	63	72	70
GT	56	58	58
Total	143	157	156

**Table 4 jof-10-00267-t004:** Carbohydrate metabolism KEGG pathway for *B. graminis* f. sp. *tritici* Wtn1.

Carbohydrate Metabolism Pathways	Gene Count
00010 Glycolysis/Gluconeogenesis	21
00020 Citrate cycle (TCA cycle)	19
00030 Pentose phosphate pathway	15
00040 Pentose and glucuronate interconversions	8
00051 Fructose and mannose metabolism	13
00052 Galactose metabolism	8
00053 Ascorbate and aldarate metabolism	1
00500 Starch and sucrose metabolism	18
00520 Amino sugar and nucleotide sugar metabolism	18
00620 Pyruvate metabolism	22
00630 Glyoxylate and dicarboxylate metabolism	17
00640 Propanoate metabolism	16
00650 Butanoate metabolism	13
00660 C5-Branched dibasic acid metabolism	6
00562 Inositol phosphate metabolism	18

KEGG: Kyoto Encyclopedia of Genes and Genomes.

**Table 5 jof-10-00267-t005:** Terpenoids and polyketide metabolism KEGG pathways for *B. graminis* f. sp. *tritici* Wtn1.

Terpenoid and Polyketide Metabolism Pathways	Gene Count
00900 Terpenoid backbone biosynthesis	16
00909 Sesquiterpenoid and triterpenoid biosynthesis	2
00906 Carotenoid biosynthesis	1
00981 Insect hormone biosynthesis	1
00908 Zeatin biosynthesis	1
00903 Limonene degradation	2
00907 Pinene, camphor, and geraniol degradation	2
01051 Biosynthesis of ansamycins	1
00523 Polyketide sugar unit biosynthesis	1
01055 Biosynthesis of vancomycin group antibiotics	1

KEGG: Kyoto Encyclopedia of Genes and Genomes.

**Table 6 jof-10-00267-t006:** Signal transduction KEGG pathways for *B. graminis* f. sp. *tritici* Wtn1.

Signal Transduction Pathways	Gene Count
02020 Two-component system	15
04010 MAPK signaling pathway	14
04013 MAPK signaling pathway—fly	11
04016 MAPK signaling pathway—plant	4
04011 MAPK signaling pathway—yeast	53
04012 ErbB signaling pathway	4
04014 Ras signaling pathway	15
04015 Rap1 signaling pathway	8
04310 Wnt signaling pathway	12
04330 Notch signaling pathway	3
04340 Hedgehog signaling pathway	4
04341 Hedgehog signaling pathway—fly	6
04350 TGF-beta signaling pathway	9
04390 Hippo signaling pathway	8
04391 Hippo signaling pathway—fly	6
04392 Hippo signaling pathway—multiple species	4
04370 VEGF signaling pathway	9
04371 Apelin signaling pathway	13
04630 JAK-STAT signaling pathway	3
04064 NF-kappa B signaling pathway	4
04668 TNF signaling pathway	3
04066 HIF-1 signaling pathway	13
04068 FoxO signaling pathway	16
04020 Calcium signaling pathway	9
04070 Phosphatidylinositol signaling system	16
04072 Phospholipase D signaling pathway	14
04071 Sphingolipid signaling pathway	17
04024 cAMP signaling pathway	9
04022 cGMP-PKG signaling pathway	9
04151 PI3K-Akt signaling pathway	23
04152 AMPK signaling pathway	21
04150 mTOR signaling pathway	38

KEGG: Kyoto Encyclopedia of Genes and Genomes.

## Data Availability

The original contributions presented in the study are included in the article/Supplementary Material, further inquiries can be directed to the corresponding authors.

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
