# Peer review of "Deciphering the Genomic Landscape and Virulence Mechanisms of the Wheat Powdery Mildew Pathogen Blumeria graminis f. sp. tritici Wtn1: Insights from Integrated Genome Assembly and Conidial Transcriptomics"

_jof, 2024, doi:10.3390/jof10040267_

Round 1
Reviewer 1 Report
Comments and Suggestions for Authors
1. When using the species name of a species not for the first time, it can be abbreviated. For example, "Blumeria graminis f. sp. tritici" can be used instead of "B. graminis f. sp. tritici".
2. In Line 118, the "-" in "scoring 7-9 on the 0-9 scale" should unify the format.
3. Units should be expressed in a standardized manner, for example, "3mm2" in line 172 and 174 should be written as "3mm2", " 25 x 103 " in line 172 should be written as "25×103" .
4. Fig.1: The annotations below the figure do not fully describe 1a,1b in the figure.
5. Fig.3B: The bar and description of tree value are absent. The capitalization of letters in the small labels in Fig 2 should be consistent. Fig 3, Fig 7 use uppercase, while Fig 4, Fig 9 use lowercase.
6. Line 173, 179, 183: "ml" should be written as "mL."
7. A space is necessary between the number and unit in Line 195 (300 μL).
8. Numbers should be written in a uniform format, For example, "1445" in line 349 should be written as "1,445".
9. The format of reference citing is not standardized in line 726, 747, 778, 819 and 884.
10. In this study, "A high-quality genome sequence from an Indian isolate of Blumeria graminis f. sp. tritici_Wtn1" in abstract. The words used to express the content should be objective, for example, "high-quality". In addition, when mention to high quality genome, the assembly quality of the genome needs to be compared with other genomes.
11. References 27 and 77, 90, 91, 92 are absent in the main text.
Author Response
Reply and rebuttal to the reviewer comments
Reviewer 1
Comment 1: When using the species name of a species not for the first time, it can be abbreviated. For example, "Blumeria graminis f. sp. tritici" can be used instead of "B. graminis f. sp. tritici".
Response: Thanks for the suggestion. We have abbreviated the species name throughout the manuscript
Comment: 2. In Line 118, the "-" in "scoring 7-9 on the 0-9 scale" should unify the format.
Response: That was an oversight. We have corrected this.
Comment: 3. Units should be expressed in a standardized manner, for example, "3mm2" in line 172 and 174 should be written as "3mm2, "25 x 103 " in line 172 should be written as "25×103"
Response: We have corrected these units; Thanks for the suggestions
Comment: 4. Fig.1: The annotations below the figure do not fully describe 1a, 1b in the figure.
Response: Thanks for pointing out the oversight. We have corrected this in the revision
Comment: 5. Fig.3B: The bar and description of tree value are absent. The capitalization of letters in the small labels in Fig 2 should be consistent. Fig 3, Fig 7 use uppercase, while Fig 4, Fig 9 use lowercase.
Response: Thanks for pointing out these oversights. We have corrected this in the revision
Comment: 6. Line 173, 179, 183: "ml" should be written as "mL."
Response: The unit is corrected in the revised draft.
Comment: 7. A space is necessary between the number and unit in Line 195 (300 μL).
Response: The typos and punctuations are revised and corrected in the revised version.
Comment: 8. Numbers should be written in a uniform format, For example, "1445" in line 349 should be written as "1,445".
Response: The number format is corrected in the revised version.
Comment: 9. The format of reference citing is not standardized in line 726, 747, 778, 819 and 884.
Response: All references are revised and corrected in the revised version.
Comment: 10. In this study, "A high-quality genome sequence from an Indian isolate of Blumeria graminis f. sp. tritici_Wtn1" in abstract. The words used to express the content should be objective, for example, "high-quality". In addition, when mention to high quality genome, the assembly quality of the genome needs to be compared with other genomes.
Response: Refer below the table for the details of genome assembly of Blumeria graminis f. sp. tritici sequenced so far. Not only the N50 value but also the numbers of contigs are clearly indicating that the genome of Wtn1 is the best one (Refer attached file for the table).
Comment: 11. References 27 and 77, 90, 91, 92 are absent in the main text.
Response: All references are corrected in the revised version, and cited.

Reviewer 2 Report
The manuscript analyses the genome of an important wheat fungal pathogen using an hydrid sequencing approach. Results are discussed taking into consideration the sate of the art.
The manuscript reports the analysis if a relevant fungal pathogen using interesting methods. Nonetheless, i believe the manuscript still needs some improvemnts prior to publication:
Minor comments:
English revision is needed. As an example: “Identified were 7707 protein……” (line 26); gene names should be italicized through the manuscript and should “forma specialis”; “no documented genome sequences of wheat powdery mildew were in India.” (line 88); mm2;
Line 273: “Comparative Genomic Analysis” should be “genome functional annotation”
Abstract:
Line 18: Define “hybrid” in this context.
Lines 22-23: the clade where Blumeria graminis f. sp. tritici Wtn1 is place, is distinct from what. Please clarify.
Lines 27-28: “orthologous proteins” and “distinct protein families (PFAMs).” Are very broad definitions. Please specify relevant orthologous or protein families.
Materials and Methods
Line 112: “Susceptible seedlings” of which species? please explain the infection assay and which are the “50 wheat genotypes” and how were they used (line140).
Lines 311-314: The procedure “To authenticate Blumeria graminis f. sp. tritici Wtn1, its genomic DNA underwent validation through the amplification of four pathogenesis-related genes: Protein Kinase C genes (PKC1 and PKC-like), Catalase (Cat1) gene, Alternative Oxidase Gene, and Integral Membrane Protein (pth) gene.” Should be described in Material and Methods.
Results
Lines 369-370: the authors describe the Analysis of Orthologous Genes, saying that “Exploring orthologous genes among Blumeria graminis f. sp. tritici Wtn1 and other genomes found 5139 common…”. Which genomes were compared?
Lines 357-359: the authors state that it is notable that there is a significant sharing of PFAMs, superfamilies (please remove comma) among the genomes that were analysed. Being of the same species I believe that this should be expected. Please explain why you think it is a “notable” feature. The same for CAZymes and orthologous genes (lines 369-372)
Tables and figures
Figures do not have enough quality for publication. Eg, figure 2 has “half” boxes and resolution is very poor.
Figure 1: Light microscopy and SEM images need a scale. Caption should include all relevant details (eg magnification, …)
Line 383: Table 6 should be table 1. Abbreviations should be described in table’s caption.
Supp Tables and Figures
Table S1: Add to the table caption the definitions used in the table (eg.: Bgt K; DWR 162; 0-3; ….). What do the dates in the table represent? The decimal point should be market with a “.”
Table S2: Define “low DNA concentration”. Below…?
Author Response
Reviewer 2
Query: Some relevant details on the methods are missing: Line 112: “Susceptible seedlings” of which species? Please explain the infection assay and which are the “50 wheat genotypes” and how were they used (line140).
Response: Selection of virulent strain (Bgt_Wtn1) based on virulence frequencies: A total of 15 Blumeria graminis f. sp. tritici (Bgt) isolates were employed to assess their virulence profiles. These isolates encompassed Bgt U13, Bgt Wpm4, Bgt Wpm17, Bgt 25, and Bgt 29 (also known as Bgt_Wtn1) collected from the SHZ (Southern Hill Zone) in the Nilgiri hills; Bgt SH1, Bgt SH4, Bgt SH9, Bgt SH11, and Bgt SH16 gathered from the NHZ (Northern Hill Zone) in Shimla, Himachal Pradesh; and Bgt KPm6, Bgt KPm10, Bgt KPm21, Bgt KPm38, and Bgt KPm50 obtained from the NWPZ (North Western Plain Zone) in Karnal, Haryana. These regions represent three agroclimatic zones where wheat powdery mildew is frequently observed
For the initial isolation and purification of these isolates, susceptible seedlings of Triticum aestivum (cv. WL711) were employed. Additionally, fifty wheat lines such as DWR 162, D 134, DWR 39, DBW 17, DDK 1029, DDK 1009, DWR 195, DPW 621-50, GW 496, GW 273, GW 18, WG 357, HUW 12, HW 517, HD 4672,, HD 2987, HDR 77, HD 2781, HD 2967, TL 2908, IWP 72, JW 3020, K 9423, MACS 2846, MP 1142, MACS 9, MACS 6273, NP 111, NW 1076, NP 165, NP 846, NARMADA195, NP 718, PBW 443, PDW 291, PBW 65, RAJ 4125, VL 802, WH 533, WH 291, WH 912, WH 157, WL711, including five near-isogenic lines of Triticum aestivum with distinct Pm genes, including Amigo (carrying Pm17; ZENG Fan-song et al., 2014), Chul Bidai (carrying Pm3b; XU Zhi et al., 2014), and Timgalen (carrying Pm6; XU Zhi et al., 2014), were introduced into seven-day-old conidia from fresh colonies (see supplementary table-ST). The virulence percentages and frequencies of Bgt isolates. A particular Bgt isolate was classified as virulent when it exhibited susceptible reactions with a maximum rating of 0- 9, falling between 7-9 on the 0–9 scale after 7–10 days of powdery mildew inoculations under controlled conditions. Following the approach suggested by Namuco et al. (1987), the data were evaluated, and the differential reactions of Bgt isolates were identified based on the average scoring value. The total number of virulent isolates, their virulence percentages, varietal efficacies, and Pm gene efficacies were computed using resistant and susceptible reactions of Indian wheat varieties and NILs. The calculation of the total number of virulent isolates, virulence percentages, varietal efficacies, and Pm gene efficacies were calculated as per the formula give by Green’s (1966).
Query: Lines 311-314: The procedure “To authenticate Blumeria graminis f. sp. tritici Wtn1, its genomic DNA underwent validation through the amplification of four pathogenesis-related genes: Protein Kinase C genes (PKC1 and PKC-like), Catalase (Cat1) gene, Alternative Oxidase Gene, and Integral Membrane Protein (pth) gene.” Should be described in Material and Methods.
Response: Thanks for the suggestion. We have incorporated the suggestion in the revision
Query: Lines 369-370: the authors describe the Analysis of Orthologous Genes, saying that “Exploring orthologous genes among Blumeria graminis f. sp. tritici Wtn1 and other genomes found 5139 common…”. Which genomes were compared?
Response: We compared the recently sequenced genome of B. graminis f. sp. tritici Wtn1, B. graminis f. sp. tritici96224 V3.16 and B. graminis f. sp. triticale THUN-12
Query: Lines 357-359: the authors state that it is notable that there is a significant sharing of PFAMs, superfamilies (please remove comma) among the genomes that were analysed. Being of the same species I believe that this should be expected. Please explain why you think it is a “notable” feature. The same for CAZymes and orthologous genes (lines 369-372)
Response: The replies are provided below
PFAMs and superfamilies: Such genome property widely associated for offering diverse protein functionality such as signalling, host adaptation, pathogenicity, and multimeric protein structure formation. Such information helpful in deep understanding of functional genome architecture and associated mechanisms. (https://doi.org/10.3390/jof8060614, https://doi.org/10.1016/j.fgb.2014.12.003)
Orthologous genes: Such genes involved in evolution of fungi, molecular phylogenetic relationship determination, and quantitative genome assembly assessment as widely used by BUSCO tool. (https://doi.org/10.3390/jof8060614 & BUSCO: https://doi.org/10.1093/molbev/msab199).
CAZyme: Play crucial role in cell wall polysaccharide breakdown, reducing host cell integrity and pathogenesis (https://bmcgenomics.biomedcentral.com/articles/10.1186/1471-2164-14-274; https://doi.org/10.3390/jof7030185).
Major comments
Comment: The manuscript analyses the genome of an important wheat fungal pathogen using an hybrid sequencing approach. Results are discussed taking into consideration the state of the art.
Response: Thanks for the positive review on our work
Comment: The manuscript reports the analysis of a relevant fungal pathogen using interesting methods. Nonetheless, i believe the manuscript still needs some improvements prior to publication
Response: We have improved the technical contents (genome assembly was redone and now it is much improved) as well as the English language.
Minor Comments
Comment: English revision is needed. As an example: “Identified were 7707 protein……” (Line 26); gene names should be italicized through the manuscript and should “forma specialis”; “no documented genome sequences of wheat powdery mildew were in India.” (Line 88); mm2;
Response: We have improved the English language thorough the text and the suggestions are incorporated.
Comment: Line 273: “Comparative Genomic Analysis” should be “genome functional annotation”
Response: Suggestion is incorporated
Abstract:
Comment: Line 18: Define “hybrid” in this context.
Response: We have suitably explained the word hybrid
Lines 22-23: the clade where Blumeria graminis f. sp. tritici Wtn1 is place is distinct from what. Please clarify.
Response: The phylogenomic analysis placed B.graminis f. sp. tritici Wtn1 in a distinct monocot-infecting clade
Lines 27-28: “orthologous proteins” and “distinct protein families (PFAMs).” Are very broad definitions. Please specify relevant orthologous or protein families.
Response: We have provided the number of OP and PFAMs.
Materials and Methods
Query: Line 112: “Susceptible seedlings” of which species? Please explain the infection assay and which are the “50 wheat genotypes” and how were they used (line140).
Response: Selection of virulent strain (Bgt_Wtn1) based on virulence frequencies: A total of 15 Blumeria graminis f. sp. tritici (Bgt) isolates were employed to assess their virulence profiles. These isolates encompassed Bgt U13, Bgt Wpm4, Bgt Wpm17, Bgt 25, and Bgt 29 (also known as Bgt_Wtn1) collected from the SHZ (Southern Hill Zone) in the Nilgiri hills; Bgt SH1, Bgt SH4, Bgt SH9, Bgt SH11, and Bgt SH16 gathered from the NHZ (Northern Hill Zone) in Shimla, Himachal Pradesh; and Bgt KPm6, Bgt KPm10, Bgt KPm21, Bgt KPm38, and Bgt KPm50 obtained from the NWPZ (North Western Plain Zone) in Karnal, Haryana. These regions represent three agroclimatic zones where wheat powdery mildew is frequently observed
For the initial isolation and purification of these isolates, susceptible seedlings of Triticum aestivum (cv. WL711) were employed. Additionally, fifty wheat lines such as DWR 162, D 134, DWR 39, DBW 17, DDK 1029, DDK 1009, DWR 195, DPW 621-50, GW 496, GW 273, GW 18, WG 357, HUW 12, HW 517, HD 4672,, HD 2987, HDR 77, HD 2781, HD 2967, TL 2908, IWP 72, JW 3020, K 9423, MACS 2846, MP 1142, MACS 9, MACS 6273, NP 111, NW 1076, NP 165, NP 846, NARMADA195, NP 718, PBW 443, PDW 291, PBW 65, RAJ 4125, VL 802, WH 533, WH 291, WH 912, WH 157, WL711, including five near-isogenic lines of Triticum aestivum with distinct Pm genes, including Amigo (carrying Pm17; ZENG Fan-song et al., 2014), Chul Bidai (carrying Pm3b; XU Zhi et al., 2014), and Timgalen (carrying Pm6; XU Zhi et al., 2014), were introduced into seven-day-old conidia from fresh colonies (see supplementary table-ST). The virulence percentages and frequencies of Bgt isolates. A particular Bgt isolate was classified as virulent when it exhibited susceptible reactions with a maximum rating of 0- 9, falling between 7-9 on the 0–9 scale after 7–10 days of powdery mildew inoculations under controlled conditions. Following the approach suggested by Namuco et al. (1987), the data were evaluated, and the differential reactions of Bgt isolates were identified based on the average scoring value. The total number of virulent isolates, their virulence percentages, varietal efficacies, and Pm gene efficacies were computed using resistant and susceptible reactions of Indian wheat varieties and NILs. The calculation of the total number of virulent isolates, virulence percentages, varietal efficacies, and Pm gene efficacies were calculated as per the formula give by Green’s (1966).
Query: Lines 311-314: The procedure “To authenticate Blumeria graminis f. sp. tritici Wtn1, its genomic DNA underwent validation through the amplification of four pathogenesis-related genes: Protein Kinase C genes (PKC1 and PKC-like), Catalase (Cat1) gene, Alternative Oxidase Gene, and Integral Membrane Protein (pth) gene.” Should be described in Material and Methods.
Response: Thanks for the suggestion. We have incorporated the suggestion in the revision
Results
Query: Lines 369-370: the authors describe the Analysis of Orthologous Genes, saying that “Exploring orthologous genes among Blumeria graminis f. sp. tritici Wtn1 and other genomes found 5139 common…”. Which genomes were compared?
Response: We compared the recently sequenced genome of B. graminis f. sp. tritici Wtn1, B. graminis f. sp. tritici96224 V3.16 and B. graminis f. sp. triticale THUN-12
Query: Lines 357-359: the authors state that it is notable that there is a significant sharing of PFAMs, superfamilies (please remove comma) among the genomes that were analysed. Being of the same species I believe that this should be expected. Please explain why you think it is a “notable” feature. The same for CAZymes and orthologous genes (lines 369-372)
PFAMs and superfamilies: Such genome property widely associated for offering diverse protein functionality such as signalling, host adaptation, pathogenicity, and multimeric protein structure formation. Such information helpful in deep understanding of functional genome architecture and associated mechanisms. (https://doi.org/10.3390/jof8060614, https://doi.org/10.1016/j.fgb.2014.12.003)
Orthologous genes: Such genes involved in evolution of fungi, molecular phylogenetic relationship determination, and quantitative genome assembly assessment as widely used by BUSCO tool. (https://doi.org/10.3390/jof8060614 & BUSCO: https://doi.org/10.1093/molbev/msab199).
CAZyme: Play crucial role in cell wall polysaccharide breakdown, reducing host cell integrity and pathogenesis (https://bmcgenomics.biomedcentral.com/articles/10.1186/1471-2164-14-274; https://doi.org/10.3390/jof7030185).
Tables and figures
Query: Figures do not have enough quality for publication. Eg, figure 2 has “half” boxes and resolution is very poor.
Response: We have improved the image quality in the revision
Figure 1: Light microscopy and SEM images need a scale. Caption should include all relevant details (eg magnification)
Response: We have inserted the scale on the image
Query: Line 383: Table 6 should be table 1. Abbreviations should be described in table’s caption.
Response: This is corrected in the revision
Supp Tables and Figures
Query: Table S1: Add to the table caption the definitions used in the table (eg. Bgt K; DWR 162; 0-3; ….). What do the dates in the table represent? The decimal point should be market with a “.”
Response: This is corrected in the revision
Query: Table S2: Define “low DNA concentration”. Below…?
Response: This is corrected in the revision; It was less than a nanogram of DNA
Reviewer 3 Report
Comments and Suggestions for Authors
This work is of high quality and of great economic importance, as wheat powdery mildew is a great evil for agriculture in many wheat-growing countries of the world. The authors have combined genomic and transcriptomic approaches to decipher the mechanisms of pathogenesis of Blumeria graminis f. sp. tritici, the pathogen causing this disease. The following are comments that need to be addressed.
1. On lines 24-25, the names of stop codons (Opal, Ochre, Amber) should be provided with the corresponding trinucleotides (TGA, TAA, and TAG, respectively).
2. The sentence located on lines 26-28 should be revised. «Identified were 7707 protein-encoding genes, 583 tRNAs, 805 effectors, 156 CAZymes, orthologous proteins, and 3180 distinct protein families (PFAMs)» confuses, because «805 effectors, 156 CAZymes, and 3180 distinct protein families (PFAMs)» are also proteins. I would suggest the following version: «Identified were 583 tRNAs and 7707 protein-encoding genes, including 805 effectors, 156 CAZymes, and 3180 distinct protein families (PFAMs)». Also, it is not clear from this sentence what the phrase "orthologous proteins" refers to, perhaps this should be clarified.
3. On line 63, “Blumeria” should be written in italics.
4. On line 136, the abbreviation “PCA” should be decoded, although it seems obvious to most readers.
5. One of the formal requirements says to enumerate the sections (e. g. “1. Introduction”, “2. Materials and Methods”) and subsections (e. g. “2.1. Virulence Profiling and Selection of Isolate”, “2.2. Selection of Powdery Mildew Pathogen for WGS”), as in the journal’s template. This will allow readers to structure what they have written. There should be no colons at the end of the (sub)section names (they are in “ Virulence Profiling and Selection of Isolate:”, “ Genomic DNA Isolation and Strain Identification:”) and dots (“Sequence Read Quality and Genome Assembly Evaluation.”). The slant, font size and boldness for sections and subsections should also be as in the mdpi template.
6. Another formal correction concerns captions. Captions to figures and tables should be made in bold black font (e. g. “Table 1”, “Figure 2”). References to them in the text should be with the full title, in a thin black font with a blue numeral (e. g. “ Table 1”, “Figure 2”). References to Supplementary materials should be made in thin black font (e. g. “ Supplemental Table S1”).
7. On lines 198-199 it is written: «The resulting PCR amplicons covering the entire barcode length were sequenced bidirectionally using the dideoxy chain termination method». Please specify the platform and reagents used for sequencing.
8. Please correct the typo on line 200: “Blast Local Alignment Search Tool” as the correct version would be “Basic Local Alignment Search Tool” and BLAST is the abbreviation.
9. As DNA isolation from such objects is a complex experimental procedure, please describe it in more detail or refer to other papers or protocols for easier reproduction of the work.
7. Please specify the program that was used to assess the quality of fastQ files. Also, please specify the versions of all bioinformatics packages used in the work (Samtools, MaSuRCA, POLCA, OrthoANI, eggNOG, DIAMOND, BWA, MEM, etc.) similar to what was done for QUAST, BUSCO, etc.
7. The ANI metric for genome comparison is often used in the case of bacteria. Only a few papers have applied it to fungi (e.g. https://onlinelibrary.wiley.com/doi/full/10.1111/jeu.12944?casa_token=MlzaNPgCp-AAAAAA%3ADE4oWUD2TmlTv8HqW2NPz5Bv-vywV1AWqU_bAOCphgHi_0DF5Pbl1c2HR-x8JHIFCtaW8lJXWi_6BWx-5w). Since fungi are eukaryotes and have a different genome structure, could you justify the need to use the same ANI values for species delimitation as in the case of bacteria.
7. According to NCBI, GCA_905067625.1 is a genome of Blumeria graminis f. sp. triticale rather than tritici. Please, make corresponding amends.
7. Figure 1 and Figure 3 should be rather moved into the Results section, as they represent the results obtained during the study, not the experimental procedures themselves (as Figure 2 represents).
7. Please, explain carefully, what Pm genes exactly are. It is currently not obvious from the text.
7. Please, replace the references on line 510 “(https://doi.org/10.1094/MPMI-08-21-0201-R)” and 587 “(https://doi.org/10.3390/jof4010039)” with the appropriate citations.
7. The gene names should be written it italics (e. g. on lines 313-314: PKC1, PKC-like, Cat1, pth; on lines 636-638 Glucose repressible protein, Oleate induced peroxisomal protein POX8, Superoxide-dismutase (SOD), Glycosidase, F-box domain-containing protein, and Catalase-peroxidase). While protein names should be written in regular font.
7. Please, tell a little about the basic mechanisms of the plant immunity. For example, that plant senses pathogen-associated molecular pattens (PAMPs) and it can launch PAMP-triggered immunity (PTI), which can be blocked by effector proteins, while effector proteins can activate some other receptors, those will activate effector-triggered immunity (ETI) and so on. If you introduce such an explanation, this would be better for understanding the role of the effectors and other proteins discussed.
7. Could you please classify transposons in the B. graminis genome (simple or complex, DNA transposons or retrotransposons) as in detailing as possible in this case and count each category.
7. Please, introduce a circular diagram, analogous to what is known for human genome (for example, Figure 6 here: https://www.nature.com/articles/s41430-021-00905-6), for B. graminis. This will make its genome composition much more demonstrative.
7. The comparison between chromosomes is very interesting, but it requires normalization per chromosome length. Could you please add a plot about gene density per n kilobases on the chromosome and SNP density per n kilobases on the chromosome? This will make the comparison less length-dependent.
7. And the final wish would be to make a scheme that summarizes the established events, genes and processes involved in B. graminis pathogenesis, no matter how many unknown places remain - it will encourage other researchers to do further work.
Author Response
Comment: This work is of high quality and of great economic importance, as wheat powdery mildew is a great evil for agriculture in many wheat-growing countries of the world. The authors have combined genomic and transcriptomic approaches to decipher the mechanisms of pathogenesis of Blumeria graminis f. sp. tritici, the pathogen causing this disease. The following are comments that need to be addressed.
Response: Thanks for the positive review on our work
Comment: 1. On lines 24-25, the names of stop codons (Opal, Ochre, Amber) should be provided with the corresponding trinucleotides (TGA, TAA, and TAG, respectively).
Reply: Suggestion incorporated
Comment: 2. The sentence located on lines 26-28 should be revised. «Identified were 7707 protein-encoding genes, 583 tRNAs, 805 effectors, 156 CAZymes, orthologous proteins, and 3180 distinct protein families (PFAMs)» confuses, because «805 effectors, 156 CAZymes, and 3180 distinct protein families (PFAMs)» are also proteins. I would suggest the following version: «Identified were 583 tRNAs and 7707 protein-encoding genes, including 805 effectors, 156 CAZymes, and 3180 distinct protein families (PFAMs)». Also, it is not clear from this sentence what the phrase "orthologous proteins" refers to, perhaps this should be clarified.
Response: We have provided the number of orthologous proteins annotated in the genome (6102 orthologous proteins)
Comment:: 3. On line 63, “Blumeria” should be written in italics.
Reply: Suggestion incorporated.
Comment: 4. On line 136, the abbreviation “PCA” should be decoded, although it seems obvious to most readers.
Reply: Suggestion incorporated.
Comment: 5. One of the formal requirements says to enumerate the sections (e. g. “1. Introduction”, “2. Materials and Methods”) and subsections (e. g. “2.1. Virulence Profiling and Selection of Isolate”, “2.2. Selection of Powdery Mildew Pathogen for WGS”), as in the journal’s template. This will allow readers to structure what they have written. There should be no colons at the end of the (sub) section names (they are in “Virulence Profiling and Selection of Isolate:”, “Genomic DNA Isolation and Strain Identification”) and dots (“Sequence Read Quality and Genome Assembly Evaluation.”). The slant, font size and boldness for sections and subsections should also be as in the mdpi template.
Reply: Suggestion incorporated.
Comment: 6. Another formal correction concerns captions. Captions to figures and tables should be made in bold black font (e. g. “Table 1”, “Figure 2”). References to them in the text should be with the full title, in a thin black font with a blue numeral (e. g. “Table 1”, “Figure 2”). References to Supplementary materials should be made in thin black font (e. g. “Supplemental Table S1”).
Reply: Suggestion incorporated.
Comment: 7. On lines 198-199 it is written: «The resulting PCR amplicons covering the entire barcode length were sequenced bidirectionally using the dideoxy chain termination method». Please specify the platform and reagents used for sequencing.
Reply: Suggestion incorporated.
Comment: 8. Please correct the typo on line 200: “Blast Local Alignment Search Tool” as the correct version would be “Basic Local Alignment Search Tool” and BLAST is the abbreviation.
Reply: Suggestion incorporated.
Comment: 9. As DNA isolation from such objects is a complex experimental procedure, please describe it in more detail or refer to other papers or protocols for easier reproduction of the work.
Reply: We have incorporated the suggestion.
Comment: 9. Please specify the program that was used to assess the quality of fastQ files. Also, please specify the versions of all bioinformatics packages used in the work (Samtools, MaSuRCA, POLCA, OrthoANI, eggNOG, DIAMOND, BWA, MEM, etc.) similar to what was done for QUAST, BUSCO, etc.
Reply: Details of each used tool version is now added in the revised draft.
Comment: 10. The ANI metric for genome comparison is often used in the case of bacteria. Only a few papers have applied it to fungi (e.g. https://onlinelibrary.wiley.com/doi/full/10.1111/jeu.12944?casa_token=MlzaNPgCp-AAAAAA%3ADE4oWUD2TmlTv8HqW2NPz5Bv-vywV1AWqU_bAOCphgHi_0DF5Pbl1c2HR-x8JHIFCtaW8lJXWi_6BWx-5w). Since fungi are eukaryotes and have a different genome structure, could you justify the need to use the same ANI values for species delimitation as in the case of bacteria?
Reply: We partly agree with the comment. Though, the following publications used to delineate the various fungal genomes at species level. https://doi.org/10.1038/s41396-020-0620-8. Another study also “proposed as a useful and powerful additional tool in the versatile toolbox of fungal taxonomy”. https://doi.org/10.3390/jof6040246.
Comment: 11. According to NCBI, GCA_905067625.1 is a genome of Blumeria graminis f. sp. triticale rather than tritici. Please, make corresponding amends.
Reply: Correction is now incorporated in revised draft.
Comment: 12. Figure 1 and Figure 3 should be rather moved into the Results section, as they represent the results obtained during the study, not the experimental procedures themselves (as Figure 2 represents).
Reply: We have incorporated the suggestion.
Comment: 13. Please, explain carefully, what Pm genes exactly are. It is currently not obvious from the text.
Reply: We have explained it in the revision.
Comment: 14. Please, replace the references on line 510 “(https://doi.org/10.1094/MPMI-08-21-0201-R)” and 587 “(https://doi.org/10.3390/jof4010039)” with the appropriate citations.
Reply: We have incorporated the suggestion.
Comment: 15. The gene names should be written it italics (e. g. on lines 313-314: PKC1, PKC-like, Cat1, pth; on lines 636-638 Glucose repressible protein, Oleate induced peroxisomal protein POX8, Superoxide-dismutase (SOD), Glycosidase, F-box domain-containing protein, and Catalase-peroxidase). While protein names should be written in regular font.
Response: Suggestion incorporated.
Comment: 16. Please, tell a little about the basic mechanisms of the plant immunity. For example, that plant senses pathogen-associated molecular pattens (PAMPs) and it can launch PAMP-triggered immunity (PTI), which can be blocked by effector proteins, while effector proteins can activate some other receptors, those will activate effector-triggered immunity (ETI) and so on. If you introduce such an explanation, this would be better for understanding the role of the effectors and other proteins discussed.
Response: Suggestion incorporated.
Comment: 17. Could you please classify transposons in the B. graminis genome (simple or complex, DNA transposons or retrotransposons) as in detailing as possible in this case and count each category.
Response: We have provided the repeat elements count and base coverage in studied genome. The separate detailed study in progress with recruiting more isolates for their genetic and geographic diversity.
Comment: 18. Please, introduce a circular diagram, analogous to what is known for human genome (for example, Figure 6 here: https://www.nature.com/articles/s41430-021-00905-6), for B. graminis. This will make its genome composition much more demonstrative.
Response: Making a circular genome appears to be challenging for us. Instead we have prepared pie chart depicting the genome contents as mentioned in the example suggested by the reviewer.
Comment: 19. The comparison between chromosomes is very interesting, but it requires normalization per chromosome length. Could you please add a plot about gene density per n kilobases on the chromosome and SNP density per n kilobases on the chromosome? This will make the comparison less length-dependent.
Response: Thank you for suggestion. Although, we could assemble the genome near about 93 scaffolds, hence for chromosome level gene density plot is currently appears little over/biased representation. On the other hand, for SNP/Variant density, we used the chromosome level reference genome, hence in revised draft MS presented the variant density/frequency in fig. no
Comment: 20. And the final wish would be to make a scheme that summarizes the established events, genes and processes involved in B. graminis pathogenesis, no matter how many unknown places remain - it will encourage other researchers to do further work.
Response: Thank you for suggestion. We have added the schematic summary of B. graminis virulence and pathogenesis.

Round 2
Reviewer 1 Report
The author of this manuscript has already made revisions according to the feedback.
The author of this manuscript has already made revisions according to the feedback.
Reviewer 3 Report
The authors have done a great job and improved the manuscript significantly by responding to all the comments. Only technical edits remain, which can be corrected during the manuscript revision process.
Technical remarks that can be corrected during the editing of the article: lines 301, 347, 713 of references should be formatted according to the journal's rules, and missing references should be added to the reference list. The date of access should be added to the Internet addresses in the text.